# Measurement report: Observation-based formaldehyde production rates and their relation to OH reactivity around the Arabian Peninsula

Dirk Dienhart[1], John N. Crowley[1], Efstratios Bourtsoukidis[2], Achim Edtbauer[1], Philipp G. Eger[1], Lisa Ernle[1], Hartwig Harder[1], Bettina Hottmann[1], Monica Martinez[1], Uwe Parchatka[1], Jean-Daniel Paris[3,2], Eva Y. Pfannerstill[1], Roland Rohloff[1], Jan Schuladen[1], Christof Stönner[1], Ivan Tadic[1], Sebastian Tauer[1], Nijing Wang[1], Jonathan Williams[1,2], Jos Lelieveld[1,2] and Horst Fischer[1]

[1]Atmospheric Chemistry Department, Max Planck Institute for Chemistry, Mainz, Germany
[2] Climate and Atmosphere Research Centre, The Cyprus Institute, Nicosia, Cyprus
[3] Laboratoire des Sciences du Climat et de l'Environnement, CEA-CNRS-UVSQ, UMR8212, IPSL, Gif-sur-Yvette, France

*Correspondence to*: Dirk Dienhart (D.Dienhart@mpic.de) or Horst Fischer (Horst.Fischer@mpic.de)

**Abstract.** Formaldehyde (HCHO) is the most abundant aldehyde in the troposphere. While its background-mixing ratio is mostly determined by the oxidation of methane, in many environments, especially in the boundary layer, HCHO can have a large variety of precursors, in particular biogenic and anthropogenic volatile organic compounds (VOCs) and their oxidation products. Here we present shipborne observations of HCHO, hydroxyl radical (OH) and OH reactivity (R(OH)), obtained during the Air Quality and Climate Change in the Arabian Basin (AQABA) campaign in summer 2017. The loss rate of HCHO was inferred from its reaction with OH, measured photolysis rates, and dry deposition. In photochemical steady-state, the HCHO loss is balanced by production via OH initiated degradation of VOCs, photolysis of oxygenated VOCs (OVOCs) and the ozonolysis of alkenes. The slope $\alpha_{eff}$ from a scatter plot of the HCHO production rate versus the product of OH and R(OH)$_{eff}$ (excluding inorganic contribution) yields the fraction of OH reactivity that contributes to HCHO production. Values of $\alpha_{eff}$ varied between less than 2 % in relatively clean air over the Arabian Sea and the southern Red Sea, and up to 32 % over the polluted Arabian Gulf (also known as Persian Gulf), signifying that polluted areas harbour a larger variety of HCHO precursors. The separation of R(OH)$_{eff}$ into individual compound classes revealed that elevated values of $\alpha_{eff}$ coincided with increased contribution of alkanes and OVOCs, with the highest reactivity of all VOCs over the Arabian Gulf.

## 1 Introduction

Formaldehyde (HCHO) is a ubiquitous trace gas that can help provide insight into the dynamical and chemical processes controlling atmospheric composition as an important source of hydroperoxyl radicals (HO$_2$) (Volkamer et al., 2010; Whalley et al., 2010; Anderson et al., 2017). The global atmospheric distribution of HCHO is dominated by in situ production during the oxidation of volatile organic compounds (VOCs) (Fortems-Cheiney et al., 2012; Anderson et al., 2017), although primary emissions from biomass burning (Akagi et al., 2011; Coggon et al., 2019; Kluge et al., 2020), vegetation (DiGangi et al., 2011),

the industry sector (Parrish et al., 2012), shipping (Marbach et al., 2009; Celik et al., 2020) and agriculture (Kaiser et al., 2015) can contribute significantly to the local HCHO abundance. HCHO is the most abundant aldehyde in the atmosphere and one of few oxygenated volatile organic compounds (OVOCs) that can be measured directly from satellites (De Smedt et al., 2008; Marbach et al., 2009; De Smedt et al., 2012; De Smedt et al., 2015; De Smedt et al., 2018, Zhu et al., 2020). In particular, the

ability to measure HCHO from satellites has instigated several studies on the relation between HCHO column densities and emissions of isoprene, one of its dominant biogenic precursors (Palmer et al., 2003; Shim et al., 2005; Millet et al., 2008; Wolfe et al., 2016). In most of the free troposphere and the remote marine boundary layer, HCHO mixing ratios are determined by methane ($CH_4$) oxidation (Ayers et al., 1997; Weller et al., 2000; Wagner et al., 2002; Anderson et al. 2017) and degradation of the oxidation products methanol ($CH_3OH$), methylhydroperoxide ($CH_3OOH$), and of other ubiquitous OVOCs like acetone

(Kormann et al., 2003; Reeves and Penkett 2003; Stickler et al., 2006; Anderson et al., 2017). In polluted areas, the oxidation of a large variety of biogenic and anthropogenic precursors contributes to HCHO production (Liu et al., 2007; DiGangi et al., 2012; Wolfe et al., 2016, Wennberg et al., 2018, Kluge et al., 2020). $HCHO/NO_2$ ratios have been used to differentiate between nitrogen oxide ($NO_x = NO + NO_2$) and VOC limited ozone production (Martin et al., 2004; Duncan et al., 2010; Schroeder et al., 2017; Tadic et al., 2020) and to infer global hydroxyl radical (OH) concentrations (Wolfe et al., 2019). If HCHO loss rates

are well defined, estimation of the concentrations of OH from HCHO mixing ratios is feasible since HCHO production is dominated by the oxidation of VOCs via OH.

Formaldehyde production in the troposphere results from reactions of VOCs with OH, ozone ($O_3$) and the nitrate radical ($NO_3$), with the oxidation processes to be dominated by OH over the day, and by $NO_3$ during night, for many trace-gases of biogenic origin. Liebmann et al. (2018) showed that the daytime loss of biogenic compounds via reaction with $NO_3$ can also be

significant in forested areas. Unsaturated hydrocarbons (e.g. ethene, isoprene and terpenes) additionally react with $O_3$ in the form of a cycloaddition to form energy-rich primary ozonides (Criegee intermediates) that rapidly fragment, releasing HCHO (Cox et al., 2020). This class of reactions is most important in relation to biogenic emissions in forested regions. Many OVOCs (e.g. alcohols, aldehydes, hydroperoxides, alkylnitrates) also produce HCHO in reactions with OH, chlorine radicals or via photolysis. Alkanes react with OH forming saturated peroxy radicals ($RO_2$) that further react with NO to form alkoxy radicals

and subsequently carbonyl compounds including HCHO. Since the oxidation of almost every VOC can produce HCHO, some with a yield greater than unity (Luecken et al., 2018), HCHO is an ideal candidate to test our understanding of VOC chemistry using zero dimensional box (Wagner et al., 2002; Fried et al., 2011) and three dimensional general circulation models (Kormann et al., 2003; Liu et al. 2007; Klippel et al., 2011; Anderson et al., 2017; Dienhart et al. 2021).

In addition to the integral of contributions by individual HCHO production pathways, the production rate of HCHO resulting

from reactions involving OH chemistry ($P_{OH}(HCHO)$) can be deduced from the OH concentration ([OH]), the HCHO yield $\alpha_i$ and the OH reactivity (R(OH)) which represents the summation of trace gases $R_i$ that react with OH with the rate coefficient $k_i$ (Liu et al. 2017; Wolfe et al., 2019):

$$P_{OH}(HCHO) = \alpha_i \cdot [OH] \cdot R(OH)_i \qquad (1)$$

with

$$R(OH) = \sum k_i \cdot R_i \qquad (2)$$

R(OH) includes reactions of OH with species like e.g. carbon monoxide (CO), sulphur dioxide ($SO_2$), nitrogen dioxide ($NO_2$) or nitrogen oxide (NO) that do not result in HCHO formation as well as reactions with VOCs like methane, alkanes, alkenes,

aromatics or OVOCs (Williams and Brune, 2015). Calculating $P_{OH}(HCHO)$ this way has the advantage that reactions of non-measured VOCs with OH will be included. Not all reactions with OH will produce HCHO, which is accounted for by the yield factor $\alpha$ (Wolfe et al., 2019), which is a summation over the HCHO yield $\alpha_i$ of each $k_i \times R_i$. For reactants that do not yield HCHO (e.g. NO, $NO_2$, $SO_2$, CO, $O_3$ and HCHO itself), $\alpha_i$ is zero. Species that yield HCHO have positive $\alpha_i$ values (e.g. $CH_4$, isoprene, etc.). Note that the individual $\alpha_i$ and thus the overall $\alpha$ can be functions of $NO_x$ (Wolfe et al., 2016). For example,

the yield of HCHO from methane oxidation depends on the fate of the initially formed methyl peroxy radical ($CH_3O_2$). At high $NO_x$ levels, $CH_3O_2$ will react with NO, subsequently forming HCHO. At low $NO_x$, $CH_3O_2$ preferentially reacts with the hydroperoxy radical ($HO_2$) forming methyl hydroperoxide ($CH_3OOH$), reducing the yield of HCHO production from $CH_4$ oxidation (Wagner et al., 2002). Since the production of HCHO is not only controlled by OH chemistry, $P_{add}(HCHO)$ represents additional sources for instance direct emissions, its production due to photolysis (e.g. $CH_3OOH$, acetaldehyde ($CH_3CHO$) and

further OVOCs) as well as the ozonolysis of alkenes (Stickler et al., 2006; Parrish et al., 2012; Anderson et al., 2017).

$$P(HCHO) = P_{OH}(HCHO) + P_{add}(HCHO) \qquad (3)$$

In photochemical steady-state (PSS), P(HCHO) is expected to be balanced by HCHO losses (L(HCHO)) and is most likely to be achieved at midday ($j_{HCHO} \sim 7 \times 10^{-5}$; R(OH + HCHO) $\sim 4 \times 10^{-5}$), when the formaldehyde lifetime is $\sim 2.5$ hours. In this study we used the PSS assumption to calculate P(HCHO) via its loss reactions:

$$P(HCHO) = L(HCHO) = \left( k_{OH+HCHO} \cdot [OH] + j_{HCHO} + \frac{v_d}{BLH} \right) \cdot [HCHO] \qquad (4)$$

The first term within brackets in Eq. 4 represents HCHO loss due to reaction with OH radicals, while the second term describes total losses due to photolysis (both the radical and molecular channel). In the boundary layer, additional loss due to dry deposition has to be considered, and depends on the deposition velocity $v_d$ and the boundary layer height (BLH), i.e. its mixing volume. Further loss due to rain-out has been neglected in this study since we did not encounter significant precipitation during

the AQABA campaign. The derived production and loss rates of HCHO can be influenced by direct emissions or by advective transport. Obvious direct emissions from ship plumes or other sources were excluded from the dataset. The potential role of transport is addressed in section 4.

In this study we use in situ observations of HCHO, OH, R(OH) and $j_{HCHO}$ together with the HCHO deposition velocity $v_d$ (calculated for 34 m above the ocean, Stickler et al., 2007) and ERA-5 meteorological data of the BLH

(https://www.ecmwf.int/en/forecasts/datasets/reanalysis-datasets/era5), obtained during a ship-cruise around the Arabian Peninsula as part of the AQABA (Air Quality and Climate in the Arabian Basin) campaign in summer 2017. From this dataset,

we calculated the loss rates of HCHO, which represent P(HCHO) during PSS conditions. Scatter plots of [OH] $\times$ R(OH) versus P(HCHO) yield the lower estimate of the formaldehyde yield $\alpha$ with respect to total OH chemistry (including $NO_x$, $SO_2$ and other non-HCHO producing reactions, Fig. S2), reflecting the transition between rather clean to highly polluted conditions, both with respect to $NO_x$ and VOCs.

$$5 \quad \alpha = \frac{P(HCHO) - P^*_{add}(HCHO)}{[OH] \cdot R(OH)} \qquad (5)$$

For further interpretation, the effective HCHO yield $\alpha_{eff}$ was determined for each region by removal of non-HCHO yielding reactions (of NO, $NO_2$, $SO_2$, CO, HCHO and $O_3$ with OH) from the OH reactivity data ($R(OH)_{eff}$, Fig. S1).

$$\alpha_{eff} = \frac{P(HCHO) - P_{add}(HCHO)}{[OH] \cdot R(OH)_{eff}} \qquad (6)$$

The major questions addressed in this study are:

10      1.   Are reactions involving OH the dominant HCHO source in the different regions around the Arabian Peninsula, or do photolysis (e.g. of oxidised organics), reactions (e.g. of unsaturated hydrocarbons) with $O_3$ or direct emission contribute significantly to the HCHO distribution? Can this method be used to identify whether the local HCHO distribution can be explained through OH oxidation only?

     2.   What is the contribution of different VOC compound classes (alkanes, alkenes, aromatics, OVOC) towards HCHO
15      production via reaction with OH?

In section 2, we give a brief outline of the AQABA cruise and the measurements performed. An investigation of the balance between HCHO production via the product of R(OH) and OH (Eq. 1) and production deduced from HCHO loss assuming PSS (Eq. 3) is given in section 3. Section 4 covers a discussion of the variation of $\alpha$ and $\alpha_{eff}$ in different regions, a summary of the results obtained is presented in section 5.

## 2 Measurements during AQABA

Measurements during AQABA took place on board the research and survey vessel *Kommandor Iona* from 25 June to 3 September 2017. The first leg from southern France to Kuwait started in La-Seyne-sur-Mer (near Toulon, France) and continued via the Mediterranean, the Suez Canal, the Red Sea, the Arabian Sea, the Gulf of Oman and the Arabian Gulf (also known as the Persian Gulf) to Kuwait. During the second leg, the vessel returned via the same route (Fig. 1).

Four laboratory containers with instrumentation for in situ and offline monitoring of a large suite of trace gases, particles and radicals were mounted on the front deck of the ship. With the exception of aerosols and radical measurements (OH and $HO_2$), air sampling was achieved from a high-flow ($10 \text{ m}^3 \text{ min}^{-1}$) cylindrical stainless steel inlet (HFI, sampling height: 5.5 m above deck, diameter: 0.2 m), placed between the containers on the front deck of the ship. Air was drawn from the center of the HFI into the air-conditioned laboratory containers using PFA (perfluoroalkoxy alkane) tubing. The inlet for OH and $HO_2$ measurements was mounted on top of a laboratory container closest to the bow.

Formaldehyde measurements were based on the Hantzsch technique using a commercial instrument (Aero-Laser, model AL4021, Garmisch-Partenkirchen, Germany). The limit of detection (LOD), determined from the reproducibility of on-board zero air measurements was between 80 and 128 pptv (influenced by wave-induced rolling of the ship) with a confidence interval of $1\sigma$. The total measurement uncertainty including line losses was 13 % (Dienhart et al., 2021).

Measurements of OH radicals were performed using laser induced fluorescence with the HORUS instrument based on laser induced fluorescence detection (Martinez et al., 2010; Hens et al., 2014, Marno et al., 2020). Typical detection limits for OH were between $1 \times 10^5$ and $5 \times 10^5$ molec $cm^{-3}$ with the total uncertainty of ~ 30 %. OH measurements below the instrumental LOD were excluded from this study. Note that an inlet pre-injector (IPI) was used to determine the OH background signal via chemical modulation (Novelli et al., 2014).

Total OH reactivity was measured using the comparative reactivity method (Sinha et al., 2008, Fuchs et al., 2017). A detailed description of the measurement technique and the results from the AQABA campaign can be found in Pfannerstill et al. (2019). The 5 min detection limit was $5.4 \text{ s}^{-1}$, derived from the $2\sigma$ standard deviation of clean air measurements over the Arabian Sea. Total uncertainty ($1\sigma$) of the OH reactivity measurements was 7–60 % with mean and average of 50 %.

Nitrogen oxides ($NO_x = NO + NO_2$) were measured with a two-channel chemiluminescence instrument CLD 790 SR (ECO Physics AG, Dürnten, Switzerland). A detailed description of the instrument during AQABA can be found in Tadic et al. (2020). The measurement uncertainty of the NO data was calculated to be 6 % at 5 min integration time and a confidence level of $1\sigma$. The LOD for the NO channel was estimated as the full width at half maximum of the frequency distribution of all zero air measurements obtained during the campaign to be 9 pptv with a confidence level of $1\sigma$. The total uncertainty of the $NO_2$ data was estimated as a conservative upper limit at 23 % as the average of the relative uncertainties of all data points obtained during the campaign. The absolute detection limit of the $NO_2$ instrument was estimated at 112 pptv (Tadic et al. 2020).

$O_3$ measurements were performed with a commercial absorption photometer (Model 202 Ozone Monitor, 2B Technologies, Boulder, Colorado). Water and particle interferences during the expedition were minimized by sampling through a Nafion tube and a Teflon filter, the overall uncertainty of the data was 2 % (Tadic et al. 2020).

CO and $CH_4$ mixing ratios were determined with a cavity ring-down spectrometer (Picarro G2401; Santa Clara, USA) with a
precision of $\leq 8$ ppbv for CO and $\leq 0.3$ ppbv for $CH_4$. The air was not dried prior to analysis and water vapor effects were corrected. The data was quality-controlled following ICOS (Integrated Carbon Observing System) standards, further details can be found in Paris et al. (2021).

Non-methane hydrocarbons (NMHCs) were measured with two coupled GC–FID systems (GC5000VOC and GC5000BTX; AMA Instruments GmbH, Germany). The GC5000VOC was used for the quantification of light hydrocarbons ($C_2$–$C_6$), while
the CG5000BTX was used for the heavier hydrocarbons and aromatics ($C_6$–$C_8$). The NMHCs have average detection limits in the range of 1–25 pptv and a total uncertainty in the range of 6–13 % (Bourtsoukidis et al., 2019).

OVOCs were detected with a PTR-ToF-MS (8000, Ionicon Analytik GmbH, Innsbruck, Austria) with a total measurement uncertainty in the range of 6–17 %, an accuracy of ~ 50 % and $3\sigma$ detection limits (derived by background measurements) of $52 \pm 26$ pptv for acetaldehyde, $22 \pm 9$ pptv for acetone and $9 \pm 6$ pptv for methyl ethyl ketone (Wang et al., 2020).

Sulfur dioxide ($SO_2$) was detected with a chemical ionization quadrupole mass spectrometer (CI-QMS) using an electrical, radio-frequency discharge ion source with a detection limit of 38 pptv and a total uncertainty of 20 % $\pm$ 23 pptv. Further details about the instrument and ionization method are described in Eger et al. (2019).

Photolysis rates for a large number of trace gases were calculated from wavelength resolved actinic flux measurements with a spectral radiometer (Metcon GmbH, Glashütten, Germany) located approximately 10 m above sea level (~ 5 m above the front
deck). The total uncertainty of the HCHO photolysis rates obtained are ~ 10 %, based on the calibration of the instrument (Bohn et al., 2008) and the data was not corrected for upwelling radiation, which is considered to be insignificant at the sea surface.

As the sampling location was in front of the ship's chimney, contamination of the measurements with the ship's own exhaust occurred when the vessel was sailing ahead of the wind. Thus, a data-filter based on relative wind direction and NO, CO and
$SO_2$ observations was used to eliminate contaminated data (including other ship exhausts and the stop in Jeddah and Kuwait). This affected mainly the measurements during the first leg of the cruise (Tadic et al., 2020).

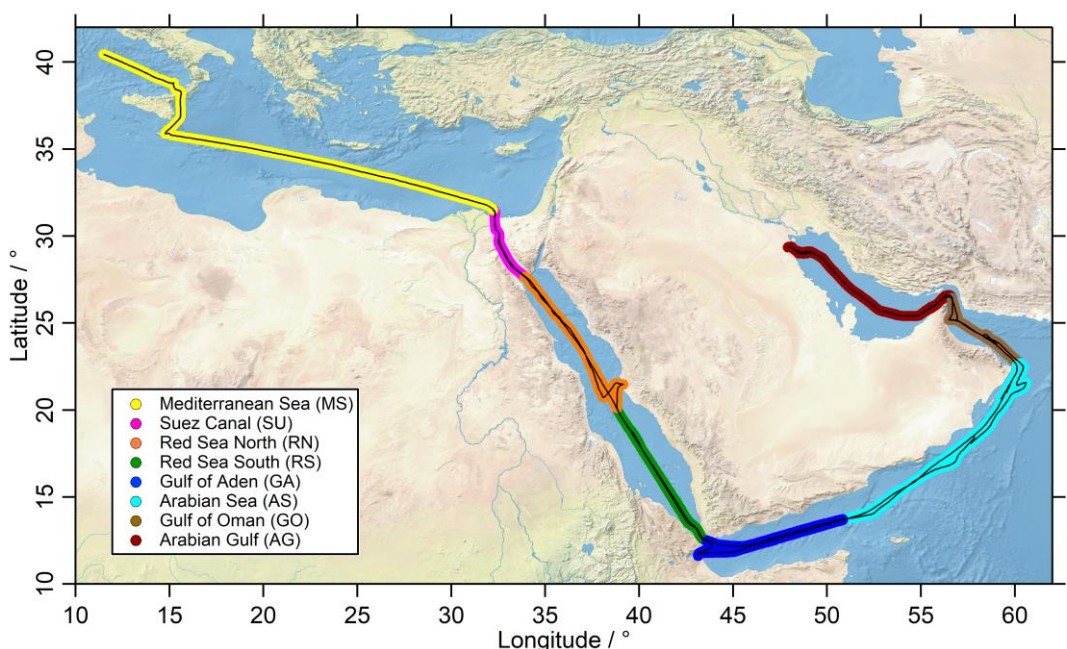

**Figure 1: The shiptrack (black) of the Kommandor Iona during the AQABA cruise subdivided into eight regions: Mediterranean sea (MS: yellow), Suez Canal (SU: pink), Read Sea North (RN: orange), Read Sea South (RS: green), Gulf of Aden (GA: blue), Arabian Sea (AS: turquoise), Gulf of Oman (GO: brown) and Arabian Gulf (AG: red).**

## 3 Results

The daytime observations of HCHO, OH, R(OH) and the photolysis rate of HCHO ($j_{HCHO}$) during the AQABA cruise are displayed in Fig. 2. Note that only daytime observations ($j_{NO2} \geq 3 \cdot 10^{-3}\,s^{-1}$) after 20 July 2017 (the start of the OH measurements) were used in this study. Formaldehyde mixing ratios varied between minimum values of approximately 0.1 ppbv over the Arabian Sea during the first leg (AS Leg 1; 20–23 July) and the Red Sea South (RS; 17–19 August) up to more than 10 ppbv over the Arabian Gulf (AG; 28 July to 5 August). The highest values were detected at the center of the Gulf, coincident with the highest ozone ($O_3$) mixing ratios > 150 ppbv during AQABA (Tadic et al., 2020). Simultaneously, elevated ethene (Bourtsoukidis et al., 2019), OVOCs (Wang et al., 2020) and organic peroxides (Dienhart et al., 2021) highlight that this area is a hotspot of air pollution. Median HCHO concentrations between 1 and 2 ppbv were measured in most of the other regions (Mediterranean Sea (MS); Red Sea North (RN); Gulf of Aden (GA), Arabian Sea Leg 2 (AS Leg 2); Gulf of Oman (OG)), while pollution events also occurred in the area around the Suez Canal (SU). In general, diurnal variations of HCHO with maximum values around local noon were observed, while the diurnal variation in clean regions like the Arabian Sea was quite small (Fig. 2).

Strong diurnal variations were observed for OH mixing ratios with noontime maximum values varying between 0.1 and 0.5 pptv in most regions. The highest noontime mixing ratios (~ 0.8 pptv) were observed over the Gulf of Aden (GA;

16 August), while air masses over the Gulf of Oman (GO) and the Mediterranean Sea (MS) also showed elevated concentrations of OH (Fig. 2). Although there is some regional variation, there is no clear trend with respect to the different locations, as for example found for HCHO mixing ratios.

Large regional variations were also found for the OH reactivity, with noontime values ranging from close to the detection limit over the Arabian Sea (AS leg 1 and 2), the Gulf of Aden (GA) and the Red Sea (RS and RN) to values of more than 30 $s^{-1}$, predominately in the polluted regions of the Arabian Gulf (AG) and the Suez Canal (SU). For a detailed discussion of the OH reactivity in different regions and its relations to VOCs and inorganic compounds see Pfannerstill et al. (2019).

Noontime-maxima in the formaldehyde photolysis rates varied between $45 \times 10^{-6}$ and $78 \times 10^{-6}$ $s^{-1}$. On most days, clear diurnal profiles were observed under clear sky conditions. Significant cloud cover was observed only on 20–22 July and 08–10 August. The boundary layer height varied between 250–1400 m (Fig. 2), with least diurnal variation in the Arabian and the Mediterranean Sea with a mean of 566 m and 780 m, respectively. The boundary layer height in other regions was more variable, with a mean value ($\pm$ 1$\sigma$) for the entire dataset (excluding the stop in Kuwait) of 750 $\pm$ 113 m.

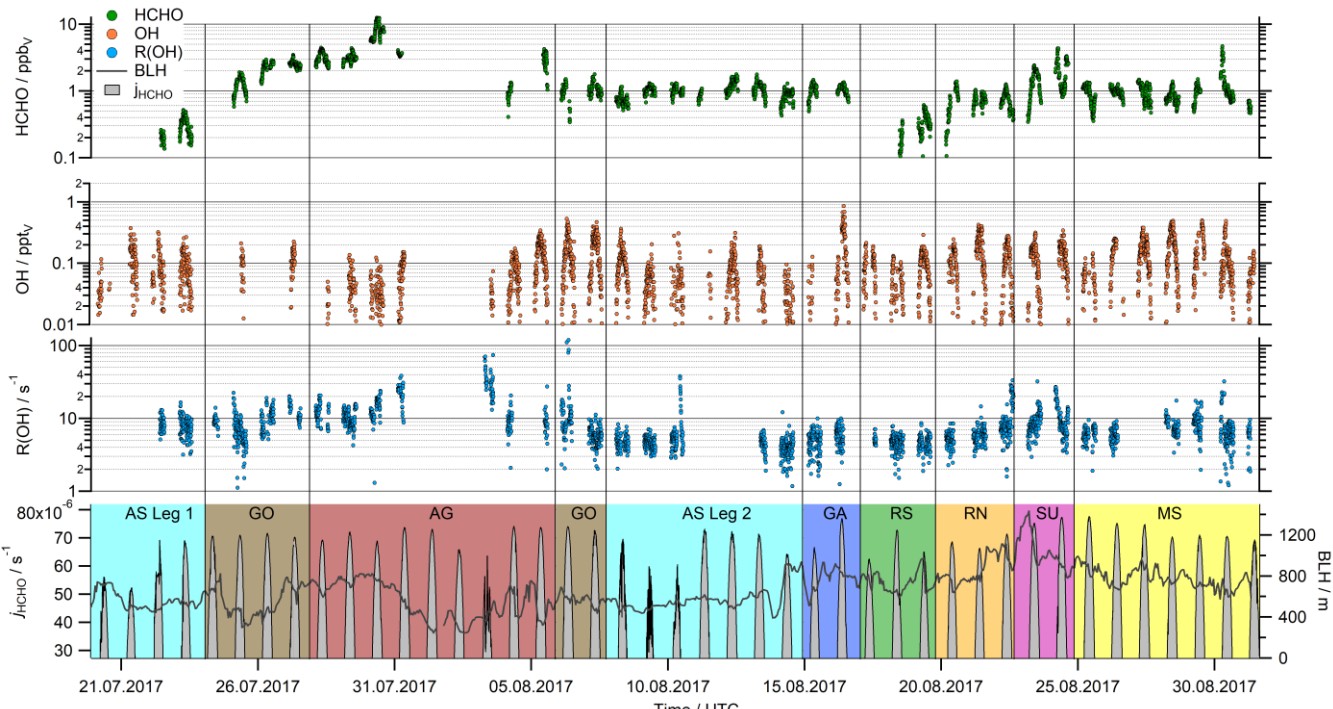

**Figure 2: Time series of HCHO, OH, R(OH) and the HCHO photolysis ($j_{HCHO}$, sum of the molecular and the radical channel) in 5 min resolution. For the boundary layer height (BLH) we used the ERA-5 meteorological reanalysis data in hourly averages, which resolves the earth's atmosphere on a 30 km grid (https://www.ecmwf.int/en/forecasts/datasets/reanalysis-datasets/era5).**

The sink of HCHO was calculated by summing the loss rate coefficients through reaction with OH, photolysis and dry deposition (Eq. 4, Fig. 3). Based on Gaussian error propagation of uncertainty in the HCHO and OH mixing ratios, $j_{HCHO}$, and

the rate coefficient $k_{OH}$ for reaction of OH with HCHO, the total uncertainties of the loss rates due to OH and $j_{HCHO}$ are 33 % and 15 %, respectively. Loss of HCHO via photolysis and reaction with OH were found to be of the same order of magnitude, with total noontime loss rates varying from 0.01 pptv s$^{-1}$ up to 1 pptv s$^{-1}$. The formaldehyde sinks due to OH and photolysis were of the order of 0.1 pptv s$^{-1}$ during noon, with the lowest values over the Arabian Sea, the Gulf of Aden (08–15 August) and the southern Red Sea (18–20 August). Significantly enhanced noontime values (> 0.2 pptv s$^{-1}$) were found for the loss through photolysis over the Arabian Gulf (28–31 August) due to the elevated HCHO mixing ratios.

The removal of HCHO by dry deposition depends on turbulent transport and also on the wind speed. Since our results for the HCHO mixing ratio are just one dimensional, the dry deposition could not be calculated directly for the AQABA dataset and was thus inferred with the literature value for the HCHO deposition velocity ($v_d$) over sea and the boundary layer height. In our analysis we used a fixed value of 0.36 ± 0.18 cm s$^{-1}$ for $v_d$ based on the findings of Stickler et al. (2007). For the BLH we used the mean value (± 1σ) of the ERA-5 results (750 ± 113 m). Assuming an uncertainty of 50 % for $v_d$ and 15 % for the BLH as conservative estimate, results in an uncertainty of 52 % for the dry deposition term.

Close to local noon, the loss rate due to dry deposition was in general below 0.1 pptv s$^{-1}$ and thus much smaller than HCHO removal by either photolysis or reaction with OH (Fig. 3). Only close to sunrise and sunset dry deposition was of similar magnitude to photochemical loss processes. The lowest panel in Fig. 3 shows the sum of all three loss processes according to Eq. 4. Based on Gaussian error propagation the estimated uncertainty of L(HCHO) is 62 %.

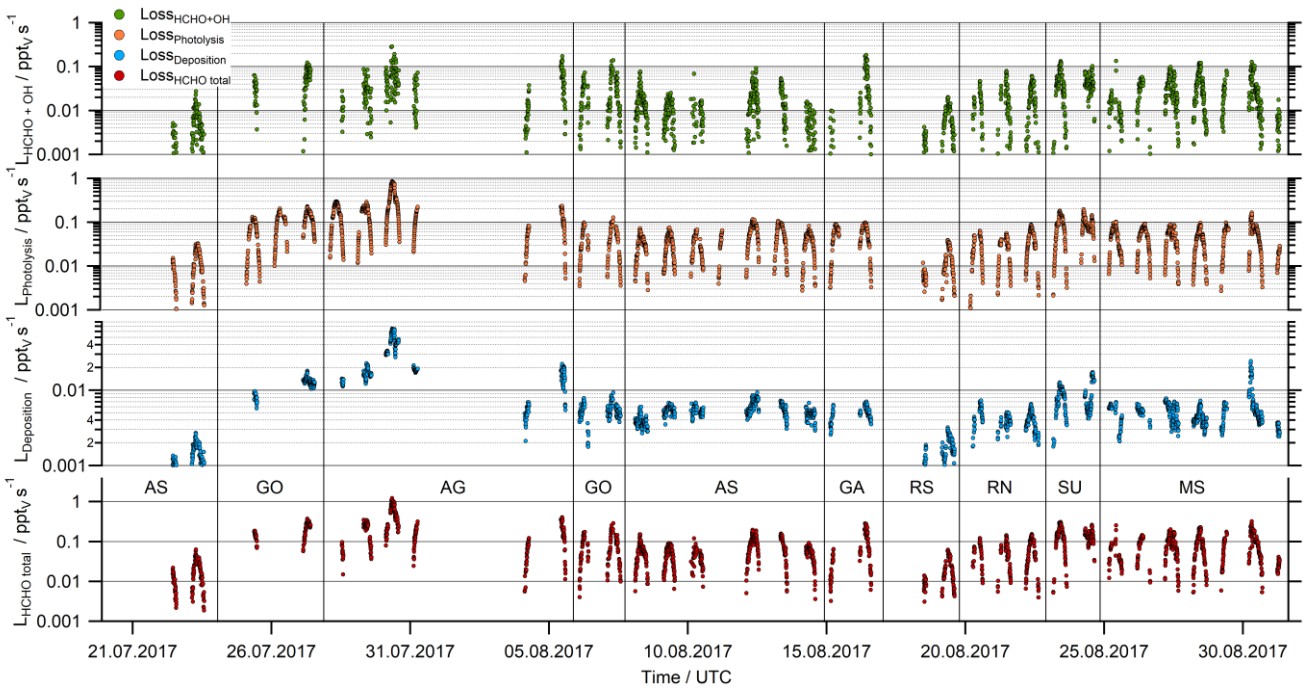

**Figure 3: HCHO loss rates due to reaction with OH, photolysis, dry deposition and the total sink as the sum of all three processes.**

In photochemical steady-state the HCHO loss and production balance, and a scatter plot of the product of the OH reactivity R(OH) ($s^{-1}$) and the OH mixing ratio (pptv) versus the HCHO production rate P(HCHO) (=$L_{HCHO\ total}$) yields the slope, i.e. the factor $\alpha$ according to Eq. 5 (Fig. 4). A compact linear relationship is expected under PSS conditions if chemical conditions do not change and the air mass is not affected by transport. While data points that do not fulfill PSS (e.g. due to direct emissions from point sources, or advection of HCHO enriched or depleted air masses) are expected to not follow the regression line, yielding additional scatter. For example, a fresh pollution plume (e.g. containing unsaturated hydrocarbons and no primary HCHO) would result in data shifted below the regression line since a high OH reactivity is expected along with low initial HCHO production, as secondary processes taking place over longer time-scales (e.g. photolysis) are involved in the HCHO production from unsaturated hydrocarbons. The dataset covers a highly polluted event in AG (30.07.2017), although the high HCHO/R(OH) ratio, as well as enhanced methanol, acetaldehyde and $O_3$ mixing ratios (Wang et al., 2020; Tadic et al., 2020) indicate rather processed air masses which could also include primary emissions of HCHO. Primary emission from point sources would likely shift data above the regression line due to the correlation with the HCHO mixing ratio (although this effect can be compensated depending on the OH concentration in emission plumes). The primary emission of HCHO cannot be accurately identified, but we removed obvious pollution events (e.g. plumes from ships or oil rigs) via covariance with elevated $NO_x$, CO and $SO_2$ mixing ratios. Furthermore, rainout can affect the relationship, because it would result in data points shifted below the regression line, although the impact can be neglected in this study as only minor precipitation occurred in AS on the 12.08.2017 during night.

In our analysis, we first subdivided the data into the smaller regions according to Fig. 1 and examined the correlation between P(HCHO) and [OH] × R(OH) for each sub-region (Fig. S2). For the Arabian Sea, the dataset has been further partitioned into the two individual legs heading towards and away from the Gulf of Oman. This is justified by the significant differences in HCHO mixing ratios and R(OH) during the two legs (AS Leg 1 and AS Leg 2) reflecting different air-mass origins. For the individual regions bivariate fits were performed according to York et al. (2004). In general, values of $R^2$ varied between 0.08 (Arabian Gulf) to 0.82 for the Gulf of Aden, while the slopes ($\alpha$) varied between less than 0.02 for the Arabian Sea during the first leg and the southern Red Sea up to 0.26 for the Arabian Gulf. The highly elevated P(HCHO) values over the Arabian Gulf (Fig. S2) also include data during a pollution event on 30.07.2017 in the center of the Gulf (Fig. 2, see also Tadic et al. 2020; Wang et al. 2020; Pfannerstill et al 2019; Bourtsoukidis et al., 2019) and thus the evaluated $\alpha$ in AG is likely influenced by primary emissions or transport of air pollution from the western coast of the Gulf. However, this pollution event seems to be representative for this unique region and thus we did not exclude the data from the study. Primary emissions of HCHO at low [OH] × R(OH) would result in data shifted towards the y-axis and thus may be partially responsible for the positive intercept on the y-axis. This would also bias the regression analysis and impact on $\alpha$. Primary emissions of HCHO at low [OH] × R(OH) would tend to increase the intercept, whereas primary emissions at high [OH] × R(OH) would bias $\alpha$ to larger values.

In general, the values of $\alpha$ represent lower limits, since the R(OH) data still includes reactions with inorganic trace-gases and other species that do not yield HCHO (e.g. CO, $NO_2$ and $SO_2$). The inorganics and CO account for ~ 10 % of the reactivity, except for the Suez Canal (~ 16 %) (see Table 1 in Pfannerstill et al., 2019). Non-zero P(HCHO) at zero [OH] × R(OH),

corresponding to the intercept b of the linear regression, can be interpreted as the result of several processes including additional chemical loss and production, dry deposition and may also be related to partial break-down of the PSS assumption when OH is low (e.g. early morning, late evening). The intercept should thus not be overinterpreted and used to e.g. calculate deposition velocities. Values for the intercept b ($P_{add}$(HCHO)) were less or equal to 0.05 pptv s$^{-1}$ in most regions, the cleanest

5  regions with respect to $NO_x$ and VOCs also showed very low values for $P_{add}$(HCHO) with approximately 0.01 pptv s$^{-1}$ over the Arabian Sea during the first leg and the southern Red Sea, indicating that HCHO production was dominated by OH chemistry. Highest values of $P_{add}$(HCHO) were found over the Arabian Gulf (0.14 pptv s$^{-1}$) and the area around the Suez Canal (0.09 pptv s$^{-1}$), where enhanced mixing ratios of unsaturated hydrocarbons and OVOCs (Wang et al., 2020) as well as elevated $O_3$ mixing ratios prevailed (Tadic et al., 2020).

10  For further analysis, we removed the highest contributors to inorganic reactivity (NO, $NO_2$, $SO_2$, $O_3$) and major non-HCHO yielding reactions (CO, HCHO) from R(OH), which resulted in so-called effective OH reactivity (R(OH)$_{eff}$). Even though the dataset was already filtered for stack emissions, $NO_x$ showed the highest contribution of these reactants (Fig. S1, Table S1), especially over the Gulf of Oman and while passing Bab-el-Mandeb (16.08.2017) and the street of Messina (30.08.2017). Over the Arabian Gulf the correction was dominated by contributions from CO, $O_3$ and HCHO. The use of R(OH)$_{eff}$ resulted in

15  reduced data coverage and an increased slope compared to when using non-adjusted R(OH) (Fig. 4, Fig. S2). The effective HCHO yield $\alpha_{eff}$ increased significantly in five sub-regions (Suez, Arabian Gulf, Gulf of Oman, Gulf of Aden, Arabian Sea Leg 2) and the plots show in general slightly reduced scatter. Values for the intercept b were not substantially affected by choice of R(OH) or R(OH)$_{eff}$ and both methods yielded similar results within their uncertainties, except for the Arabian Sea during the second leg, where the intercept decreased to similar values of the Mediterranean Sea when R(OH)$_{eff}$ was used.

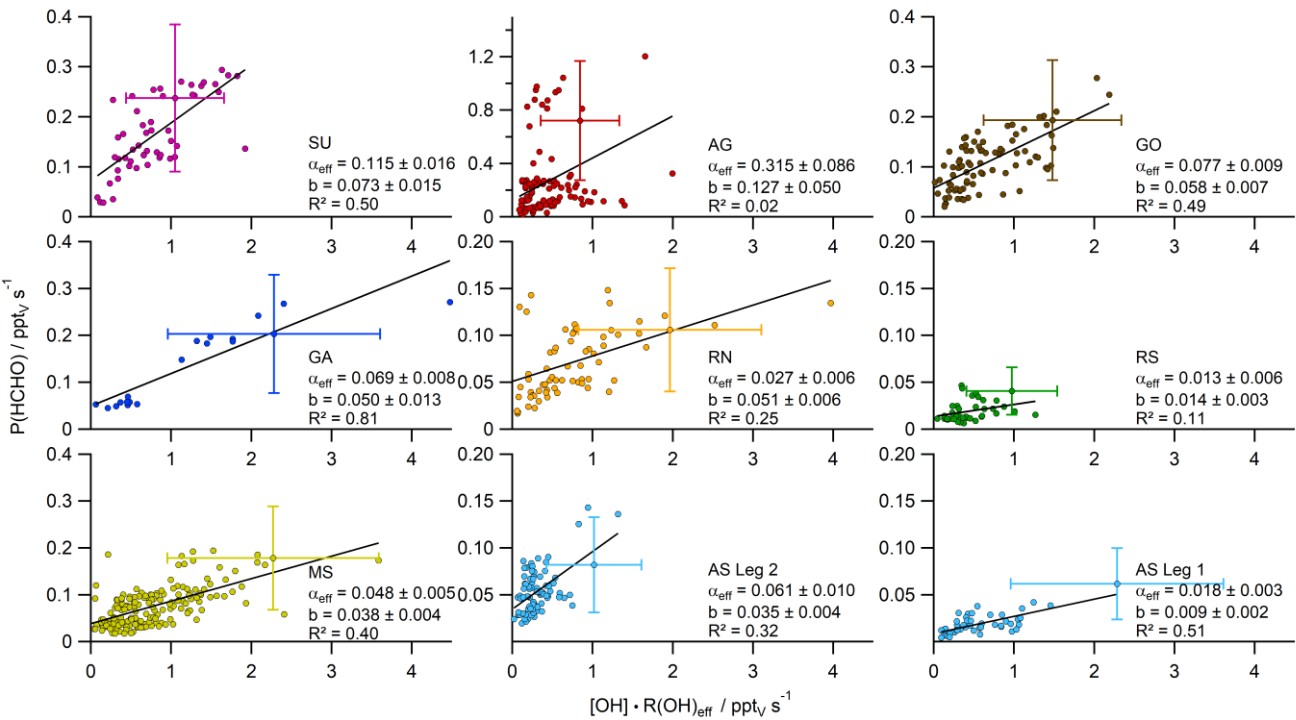

**Figure 4: Scatter plots with bivariate fits (York et al., 2004) of the product [OH] × R(OH)$_{eff}$ (± 58 %) versus the HCHO production rate P(HCHO) (± 62 %), subdivided into the different regions probed during the AQABA cruise. The slope of the respective regression represents the HCHO yield $\alpha_{eff}$, while the intercept can be interpreted as additional HCHO sources not related to OH chemistry (P$_{add}$(HCHO)).**

## 4 Discussion

The values of [OH] × R(OH)$_{eff}$ vary between practically zero and more than 4 pptv s$^{-1}$ with maximum values over the northern Red Sea, the Mediterranean and the Gulf of Aden, while most data represents values ≤ 3 pptv s$^{-1}$. According to Eq. 4, formaldehyde loss L(HCHO) is the product of the HCHO mixing ratio and the loss rates due to reaction with OH, photolysis and dry deposition. In photochemical steady-state (typically lifetime of HCHO ~ 2.5 h), formaldehyde loss is equal to its production P(HCHO) (Eq. 4). P(HCHO) varied between 0 and approximately 1.2 pptv s$^{-1}$, with values above 0.4 pptv s$^{-1}$ only detected over the Arabian Gulf, which were likely influenced by air pollution transported from the west coast of the Gulf and Kuwait, where oil and gas industries are numerous (Pfannerstill et al., 2019). Maximum P(HCHO) was determined at the center of the Arabian Gulf during the first leg, where elevated mixing ratios of unsaturated hydrocarbons, OVOCs and O$_3$ were observed (Bourtsoukidis et al. 2019, Wang et al. 2020, Tadic et al. 2020). This air mass was characterized by rather low OH, slightly elevated OH reactivity and the highest HCHO mixing ratios during AQABA (30.07.2017, Fig. 2), which resulted in data points shifted towards the y-axis (Fig. 4). We nevertheless decided to include the data as it represents highly polluted air

within the center of this unique region. Minimum P(HCHO) values were determined over the southern Red Sea and the Arabian Sea during the first leg ($\leq 0.05$ pptv $s^{-1}$). These results also highlight the limits of the method used in this study, since primary emissions and transport can significantly affect the local HCHO distribution. Furthermore, some HCHO yielding reactions require oxidation (not only via OH) and subsequent photolysis steps of rather long-lived oxidation products before releasing

HCHO and are thus temporarily decoupled from measurements of OH and OH reactivity.

In general, compact correlations between P(HCHO) and [OH] $\times$ R(OH) were observed over the Gulf of Aden, the Mediterranean Sea and the Arabian Sea during Leg 1 as indicated by $R^2$ values between 0.56 and 0.82 (Fig. S2). Lower $R^2$ (less than 0.35) were found for the Suez Canal, the Gulf of Oman, the Red Sea South and North, the Arabian Sea during the second leg and the Arabian Gulf. In the area around the Suez Canal the observations indicate a change in chemical composition,

as data points are shifted to the left and to the right of the bivariate fit. Especially the Suez Canal is prominent for its immense ship traffic (Celik et al., 2020) as well as oil and gas exploration (Bourtsoukidis et al., 2019), thus it is likely that our results interfere with primary HCHO sources or transport of air pollution in this region. Wang et al. (2020) also highlighted a biomass burning plume in this area, which was excluded from this dataset as it occurred in the early morning and thus a steady-state analysis is not appropriate. Removal of non-HCHO yielding reactions from the OH reactivity (R(OH)$_{eff}$) reduced the overall

scatter significantly, which resulted in increased $R^2$ values for the Suez Canal and the Arabian Sea during leg 2 (Fig. 4, S2). The still rather low $R^2$ for the Arabian Sea during leg 2 results from a cluster of data points at low values on both the x- and the y-axis. In this sub-region there was little variation in chemical composition as air masses (during the Indian summer monsoon) originated from the east coast of Africa (Edtbauer et al., 2020, Tegtmeier et al., 2020). In contrast, the results for the Arabian Gulf cover the highest variation of HCHO and VOCs (Bourtsoukidis et al., 2019; Wang et al., 2020) and are

characterized by substantial changes in chemical composition, although obvious point sources (e.g. ship exhausts) with enhanced NO$_x$, CO, SO$_2$ mixing ratios were excluded from the entire study.

Besides individual data points that do not follow the regression line, the intercept b of the bivariate fit can be interpreted based on Eq. 5 either as an additional loss of HCHO (e.g. wash out, deposition) or as a region wide attribution of additional HCHO sources not related to OH chemistry (P$_{add}$(HCHO), Fig 4). As mentioned above, the effect of wash-out should be negligible,

since we only experienced a short rain event on the 12.08.2017 during the night. Using literature values for the deposition velocity in this study (Stickler et al., 2007), we calculate that dry deposition can account for only 8–19 % of the intercept. The deposition velocity would have to be underestimated by a factor of five or more or the BLH would have to be overestimated by a similar factor (or a combination of both). While the elevated values representing the Arabian Gulf were influenced by the pollution event encountered at the center of the Gulf, the other regions representing enhanced concentrations of air pollution

(SU, GO, RN) also show significantly elevated intercepts within their uncertainties (0.051–0.127 pptv $s^{-1}$) compared to the relatively clean regions (AS, MS, RS), where we determined smaller intercepts b (0.009–0.038 pptv $s^{-1}$). Additional HCHO production via the ozonolysis of alkenes, indicated by elevated levels of O$_3$ (Tadic et al. 2020) and ethene (Bourtsoukidis et al., 2019), seems very likely in AG and SU. Elevated levels of OVOCs (Wang et al., 2020) and CH$_3$OOH (Dienhart et al.,

2021) were also detected in these regions and thus it is more plausible to use the intercept b as an indicator for additional HCHO production ($P_{add}$(HCHO)) not related to OH chemistry, although it seems not feasible to use for quantification.

Large variability was observed for the slope m of the regression line for the different regions. As mentioned above, this slope can be equated to the yield $\alpha$, quantifying the fraction of OH reactivity that results in HCHO production. Inorganic species and

CO made up ~ 10 % of the reactivity in most regions and 16 % over the Suez Canal (Pfannerstill et al., 2019) and thus the values for $\alpha$ are lower limits when R(OH) is used. Here $\alpha$ varies between 0.014 over the southern Red Sea and 0.258 over the Arabian Gulf, indicating a lower limit of about 1 and 25 % of R(OH) contributing to HCHO production respectively. R(OH)$_{eff}$ was utilized to examine whether $\alpha_{eff}$ would increase drastically compared to $\alpha$ (Fig. 4, S2). By comparing the scatter plots in SU and AS Leg 2 loss of data points is noticeable due to lack of either $NO_x$, $SO_2$, CO or $O_3$ measurements and thus we chose

to present both R(OH) and R(OH)$_{eff}$ in this study. The use of R(OH)$_{eff}$ reduced the span on the x-axis as expected significantly, which resulted in increased values for $\alpha_{eff}$ compared to $\alpha$ in five out of the nine sub-regions. The largest increase was determined for the Arabian Gulf (0.315), the Suez Canal (0.115) and the Gulf of Oman (0.077), all of which were rather polluted. However, the determined $\alpha_{eff}$ did not change significantly in most regions, which is also the case for the lowest values determined for the Arabian Sea (0.018) and the southern Red Sea (0.013). In these rather clean regions a low contribution of inorganic reactivity

is expected, which was proven (Fig. S1). Removal of non-HCHO yielding reactions from total OH reactivity indicates that between 1 and 32 % of R(OH)$_{eff}$ contribute to HCHO production. To put these results into perspective, Pfannerstill et al. (2019) compared the R(OH) measurements with the calculated reactivity from all measured species. In general, summation of the measured trace gases resulted in a contribution of unattributed OH reactivity between ~ 20 % over the Arabian Gulf and up to 55 % over the Gulf of Aden.

We expect HCHO to be produced from methane ($CH_4$), non-methane hydrocarbons such as alkanes and alkenes, OVOCs and to a lesser extend from aromatic hydrocarbons (Wagner et al., 2002). For further analysis, we have recalculated mean OH reactivities ($\pm$ 1$\sigma$ standard deviation) of the individual substance classes (R(OH)$_x$, e.g. alkanes, alkenes, OVOCs, aromatics) based on the findings of Pfannerstill et al. (2019). We assume that the unmeasured VOCs occurred at levels proportional to the measured compounds of a certain compound class, so that these values represent general trends for the VOC oxidation of

the subdivided regions. Please note that we removed the reaction of HCHO with OH from the OVOC class, which is usually included when presenting speciated reactivity (Table S2, see also Table 1 in Pfannerstill et al., 2019). The ratio of R(OH)$_x$/R(OH)$_{eff}$ then represents the relative contribution of a certain compound class to the regional so-called effective reactivity, which reflects the measured VOC oxidation plus unattributed reactivity. Alkanes (including $CH_4$) contributed on average 3–10 %, alkenes 2–10 %, OVOCs 8–33 % and aromatics 1–9 % to regional R(OH)$_{eff}$ (Table 1, Fig. 5). All sulfur-

containing VOCs together contributed less than 1 % to the total R(OH)$_{eff}$ and were thus neglected. Elevated contributions of alkanes to R(OH)$_{eff}$ were found in SU and AG, while their contribution in the remaining regions was rather constant (~ 5 %) and dominated by the oxidation of methane. Lowest relative contributions of the other compound classes were generally found over the Mediterranean Sea and the Arabian Sea, where also the lowest total OH reactivity has been detected (Table 1; see also Fig. 1 Pfannerstill et al., 2019). Highest relative contributions were observed over the Arabian Gulf, with alkanes and alkenes

each contributing ~ 10 % and OVOCs ~ 33 % to R(OH)$_{eff}$. The largest relative contribution of aromatic compounds was found over the area near the Suez Canal with ~ 9 %. The sum of alkanes, alkenes, OVOCs and aromatic compounds on average contributed 19 % over the Mediterranean Sea, 32 % over the Suez Canal, 31 % over the northern Red Sea, 37 % over the southern Red Sea, 23 % over the Gulf of Aden, 16 % on the first leg over the Arabian Sea, 38 % over the Gulf of Oman, 62 % over the Arabian Gulf and 23 % on the second leg over the Arabian Sea to R(OH)$_{eff}$.

**Table 1: Average speciated reactivity (R(OH)$_x$ ± 1σ standard deviation) of certain compound classes (based on the results of Pfannerstill et al., 2019), R(OH)$_{eff}$ and $α_{eff}$ (± uncertainty of the slope of the bivariate fit). For NO$_x$ the median values are listed instead of the means.**

| | R(OH)$_{alkanes}$ / s$^{-1}$ | R(OH)$_{alkenes}$ / s$^{-1}$ | R(OH)$_{OVOCS}$ / s$^{-1}$ | R(OH)$_{aromatics}$ / s$^{-1}$ | R(OH)$_{eff}$ / s$^{-1}$ | $α_{eff}$ | NO$_x$ / ppb$_v$ |
|---|---|---|---|---|---|---|---|
| **MS** | 0.323 ± 0.131 | 0.117 ± 0.179 | 0.656 ± 0.283 | 0.105 ± 0.143 | 6.412 ± 2.854 | 0.048 ± 0.005 | 0.275 ± 2.024 |
| **SU** | 0.683 ± 0.546 | 0.765 ± 1.025 | 1.118 ± 0.870 | 0.699 ± 1.178 | 10.17 ± 5.174 | 0.115 ± 0.016 | 4.806 ± 8.014 |
| **RN** | 0.449 ± 0.187 | 0.562 ± 0.522 | 1.110 ± 0.488 | 0.386 ± 0.410 | 8.192 ± 3.251 | 0.027 ± 0.006 | 0.752 ± 5.648 |
| **RS** | 0.310 ± 0.071 | 0.781 ± 0.729 | 1.171 ± 0.444 | 0.670 ± 0.644 | 7.840 ± 4.138 | 0.013 ± 0.006 | 0.232 ± 1.208 |
| **GA** | 0.290 ± 0.039 | 0.290 ± 0.372 | 0.861 ± 0.198 | 0.262 ± 0.344 | 7.502 ± 2.999 | 0.069 ± 0.008 | 0.604 ± 3.500 |
| **AS Leg 1** | 0.270 ± 0.001 | 0.174 ± 0.073 | 0.658 ± 0.046 | 0.114 ± 0.026 | 7.881 ± 1.872 | 0.018 ± 0.003 | 0.212 ± 0.192 |
| **GO** | 0.366 ± 0.100 | 0.421 ± 0.309 | 1.504 ± 0.280 | 0.427 ± 0.334 | 7.168 ± 3.548 | 0.077 ± 0.009 | 2.596 ± 4.850 |
| **AG** | 1.198 ± 1.195 | 1.208 ± 1.190 | 3.853 ± 2.438 | 0.852 ± 0.843 | 11.57 ± 5.976 | 0.315 ± 0.086 | 1.207 ± 3.997 |
| **AS Leg 2** | 0.274 ± 0.016 | 0.136 ± 0.075 | 0.476 ± 0.087 | 0.092 ± 0.061 | 4.308 ± 0.711 | 0.061 ± 0.010 | 0.155 ± 1.838 |

In general, the sensitivity of HCHO production to different VOCs results from both the per molecule yield (amount of HCHO produced per VOC molecule lost) and the total abundance (or anthropogenic emissions) of the VOC. Luecken et al. (2018) showed for the continental United States, that the per-molecule sensitivity of HCHO is highest for anthropogenic emissions of alkenes and comparable within the uncertainties for alkanes, OVOCs and aromatics. Methane oxidation via OH and other oxidants (e.g. chlorine, Cl) is expected to be the main source of HCHO in the remote marine boundary layer, via the production of methylperoxy radicals (CH$_3$O$_2$). The fate of these CH$_3$O$_2$ radicals determines the main production pathway of HCHO, as they can react to other species e.g. methyl hydroperoxide (CH$_3$OOH) or methanol (CH$_3$OH) under low-NO$_x$ conditions (Anderson et al., 2017). The determining step to form HCHO is the reaction of CH$_3$O$_2$ with NO, thus it is expected that HCHO production is suppressed in very clean regimes, since these favor the recombination of CH$_3$O$_2$ and the reaction with HO$_2$. Emission of rather short-lived alkenes (e.g. ethene, isoprene) can significantly enhance HCHO production via reaction with O$_3$ or OH and the subsequent formation of OVOCs (e.g. acetone, acetaldehyde). The release of HCHO from these secondary products not only depends on their chemical structure, but also on additional photolysis steps involved. While methane oxidation dominates the HCHO production in the MBL, isoprene is expected to be the main precursor over continents for near-

surface HCHO. The results of Wolfe et al. (2016) also highlight that the HCHO yield from isoprene oxidation is a nonlinear function of $NO_x$.

As indicated above, the yield of HCHO from $OH \times R(OH)_{eff}$ is expected to depend on the composition of $R(OH)_{eff}$ with respect to alkanes, alkenes, OVOCs and aromatic compounds and also on the presence of $NO_x$. Mixing ratios of biogenic hydrocarbons, in particular isoprene, were very low during AQABA with the exception of dimethylsulfide (Edtbauer et al., 2020). In clean MBL environments, emissions of alkenes and aromatics are less relevant and thus the distribution of HCHO should be primarily controlled via variability of $HO_x (= OH + HO_2)$ and the presence of $NO_x (> 0.1\ ppbv)$, while we expect the HCHO budget to be highly complex under more polluted conditions e.g. for the area around Suez Canal and over the Arabian Gulf.

In order to investigate if the change in chemical composition along the different regions correlates with the determined HCHO yield, Fig. 5 shows scatter plots of the HCHO yield $\alpha_{eff}$ versus the measured OH reactivity towards certain compound classes (alkanes, alkenes, OVOCs, aromatics) as the contribution to total OH reactivity without inorganic reactions (e.g. $R(OH)_{alkanes}/R(OH)_{eff}$). We also color-coded the z-axis with the median of measured $NO_x$ as an indicator for air pollution and in order to identify the regions more easily.

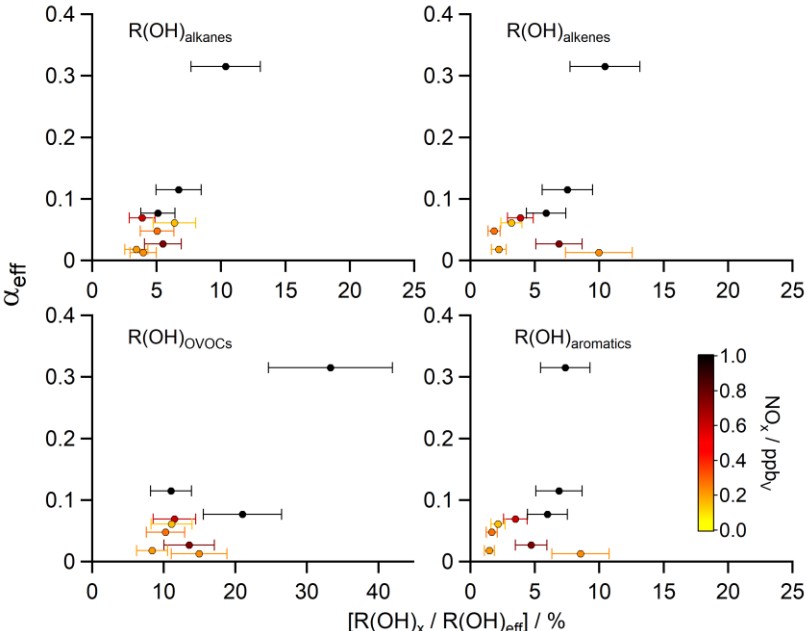

Figure 5: Scatter plots of the contribution of certain substance classes $R(OH)_x$ (e.g. alkanes, alkenes, OVOCs, aromatics) to the total OH reactivity without the contribution of inorganic reactions ($[R(OH)_x/R(OH)_{eff}]$) versus the HCHO yield $\alpha_{eff}$ (calculated with $R(OH)_{eff}$ via Eq. 6). The z-axis was color-coded with the measured $NO_x$ median values, representative for the nine different regions during AQABA (Table 1).

The relative contribution of alkanes to $R(OH)_{eff}$ shows little variation at ~ 4–5 % except over the area near the Suez Canal (~ 7 %) and over the Arabian Gulf (~ 10 %), where methane (Paris et al., 2021) and a variety of NMHCs (Bourtsoukidis et al., 2019) showed significantly elevated mixing ratios. Over the Arabian Gulf, these were likely caused by emissions from the oil

and gas industry, indicated by the ratio between i-pentane and n-pentane of $0.93 \pm 0.03$ ppbv ppbv$^{-1}$ while Bourtsoukidis et al. (2019) found a ratio of $1.71 \pm 0.06$ ppbv ppbv$^{-1}$ over the Suez Canal, which is representative for ship emissions. From visual inspection, a slight increase of the relative reactivity of alkanes lead to a small, but significant increase of $\alpha_{eff}$ with simultaneously elevated values of NO$_x$. Alkenes contributed on average between ~ 2–10 % to R(OH)$_{eff}$, with the highest values

over the Arabian Gulf, the southern Red Sea and the Suez Canal. Interestingly, a high reactivity of alkenes was detected over the southern Red Sea, which was similar to the enhanced values over the Suez Canal (Table 1), although with a higher relative contribution. The NO$_x$ measurements indicate rather clean air over the southern Red Sea (median 0.23 ppbv), where we experienced mostly winds coming from the west (Eritrea, Sudan). As a first result, a higher reactivity of alkenes not necessarily correlated with an increase of $\alpha_{eff}$, which can be partly explained by the fact that release of HCHO from alkene oxidation

depends on photolysis steps and other reactions involved, and our method does not take into account chemical aging. Additionally, the availability of other oxidants (e.g. O$_3$) was higher over the Suez Canal and the Arabian Gulf, where the highest amounts of O$_3$ were detected during AQABA (Tadic et al., 2020). This likely interferes with the presented results and could lead to an increase of P$_{add}$(HCHO). In general, emissions of alkenes favor HCHO production, which was also mentioned by Luecken et al. (2018), who found a high sensitivity of HCHO to anthropogenic and biogenic emissions of alkenes with

isoprene as the main contributor in summer. We therefore emphasize the limits of our method, as it only takes into account the `immediate yield` of HCHO. OVOCs showed overall the highest contribution to R(OH)$_{eff}$ with ~ 8–33 % during AQABA and elevated R(OH)$_{OVOCs}$ generally resulted in an increased $\alpha_{eff}$, with the highest contribution over the Arabian Gulf, the Gulf of Oman and the Suez Canal. The production of HCHO through OVOCs strongly depends on the composition and amounts of anthropogenic emissions. Some C$_1$-C$_3$ OVOCs (e.g. methanol, acetaldehyde, acetone, glyoxal) are expected to have a strong

influence on the local HCHO distribution depending on their mixing ratio, although this effect is minor in remote marine boundary layer conditions with least amounts of anthropogenic contributions (Anderson et al., 2017). High concentrations of acetaldehyde and acetone over the Arabian Gulf (Wang et al. 2020) occurred with a simultaneous increase of HCHO and highlight this region as unique spot of air pollution, which is reflected by the highest $\alpha_{eff}$ and R(OH)$_{OVOCs}$. R(OH)$_{aromatics}$ showed in general the lowest contribution to R(OH)$_{eff}$ with 1 – 9 % with again highest contribution over the Arabian Gulf, the southern

Red Sea and the Suez Canal. Luecken et al. (2018) showed that HCHO concentrations over the United States had about the same sensitivity towards aromatic and alkane emissions. Aromatic compounds do not necessarily release HCHO during their oxidation, but generally can be useful to identify complex anthropogenic emissions e.g. ship emissions or industry. These results overall underscore the elevated levels and complexity of air pollution detected over the Arabian Gulf and the Suez Canal. Also, an absolute increase in R(OH)$_x$ is mostly accompanied by an increase of $\alpha_{eff}$, although the majority of data points

is scattered at low $\alpha_{eff}$ and there is no clear correlation with R(OH)$_{aromatics}$ and R(OH)$_{alkenes}$. Our results are not directly comparable to the findings by Luecken et al. (2018), since in their analysis reactions with O$_3$ (alkenes) and photolysis of HCHO precursors (OVOCs) were included. These sources of HCHO are not represented by R(OH)$_x$ used in our study and additionally our method would be more valuable by 'following' an air mass, to account for air mass aging.

The yield of HCHO can also be interpreted as a function of $NO_x$ levels if there are processes (at low NO) that lead to formation of e.g. peroxides rather than HCHO (Wolfe et al., 2016), this occurs when the recombination of $CH_3O_2$ radicals (or $CH_3O_2 + HO_2$) is prioritized instead of the reaction with NO to form the methoxy radical $CH_3O$, which rapidly reacts with oxygen and produces HCHO (Stickler et al., 2006). NO also directly affects the availability of OH, first of all as a sink by the formation of nitrous acid (HONO). HONO rapidly regenerates NO and OH by photolysis during the day, which was identified as a major source of daytime background $NO_x$ during AQABA (Friedrich et al., 2021). Additionally, NO enhances the conversion of $HO_2$ radicals to OH and thus accelerates VOC oxidation (Wolfe et al., 2016). Valin et al. (2016) found that in isoprene rich regions, the influence of $NO_x$ on HCHO production is primarily due to its feedback on the production of OH. Wolfe et al. (2016) showed that chemical link between HCHO and isoprene is a strong, non-linear function of $NO_x$. They demonstrated an increase of the prompt yield of HCHO by a factor of 3 (from 0.3 to 0.9 ppbv ppbv$^{-1}$) over the range of observed $NO_x$ (0.1–2 ppbv), while background HCHO increased by a factor of 2 (from 1.6 to 3.3 ppbv ppbv$^{-1}$). We found a similar trend, as the increased reactivity of alkenes in the southern Red Sea at rather low $NO_x$ (median 0.23 ppbv) was not accompanied by enhanced HCHO mixing ratios, although high values of $\alpha_{eff}$ coincided with elevated $NO_x$. Please note that our method does not account for chemical aging as already mentioned above. In general, the impact of $NO_x$ on HCHO production can be reduced to two factors: radical recycling and the termination of $RO_2$ radicals (Wolfe et al., 2016). The fact that very clean background MBL air was rarely sampled during AQABA (lowest median $NO_x$ of 0.16 ppbv over AS Leg 2) suggests that we hardly achieved conditions, in which the production of peroxides through the reaction of $CH_3O_2$ with $HO_2$ would dominate the production of HCHO through $CH_3O_2 + NO$. Tadic et al. (2020) also showed that most regions were in the $O_3$ production regime (signaling sufficient NO), which indicates that the production of HCHO was more likely limited by the abundance of oxidants and reactive VOCs rather than $NO_x$.

## 5 Summary and conclusions

In-situ observations of HCHO and its sinks due to reaction with OH, photolysis and dry deposition to the ocean surface have been compared to its production from the reaction of OH with VOCs by using the total OH reactivity (R(OH)) and measured OH concentrations during the AQABA ship campaign around the Arabian Peninsula. Large variation of HCHO mixing ratios about more than an order of magnitude with maximum values (up to ~ 12 ppbv) in the center of the Arabian Gulf highlight the region as a hotspot of photochemical air pollution with elevated mixing ratios of several VOCs and high amounts of ozone ($\geq$ 150 ppbv, Tadic et al., 2020). Lowest concentrations of HCHO and its precursors identified the Arabian Sea as the cleanest region of the measurement campaign, related to the stable winds of the Indian summer monsoon, originating near the east coast of Africa (Edtbauer et al., 2020). Rather clean air (from a $NO_x$ perspective) was also detected over the southern Red Sea, which showed enhanced alkene and aromatic compound concentrations. In photochemical steady-state, compact relationships between HCHO production and loss were found for some regions along the ship track. Excluding inorganic reactivity and non-HCHO yielding reactions from total R(OH) slightly improved the correlation between production and loss of HCHO. The

region-wide yield of HCHO per reacting OH ($\alpha_{eff}$) differs along the various chemical regimes encountered with lowest values (less than 2 %) deduced over the Arabian Sea during the first leg and the southern Red Sea, and highest values (~ 32 %) over the Arabian Gulf. In most regions less than 10 % of R(OH)$_{eff}$ contributed to HCHO production. The separation of R(OH)$_{eff}$ into certain compound classes showed that OVOCs had the highest overall contribution to OH reactivity and were most variable. In general, elevated values of $\alpha_{eff}$ coincided with elevated contribution of alkanes and OVOCs, with highest reactivity of all compound classes in the Arabian Gulf. The increased reactivity of alkenes and aromatic compounds in the southern Red Sea at rather low NO$_x$ (median 0.23 ppbv) was not accompanied by an elevated $\alpha_{eff}$. A clear dependence of $\alpha_{eff}$ on NO$_x$ could not be identified, although highest values of $\alpha_{eff}$ coincided with elevated NO$_x$. Future studies on the HCHO budget around the Arabian Peninsula would likely benefit from longer stationary measurements e.g. to identify the regularity of highly polluted events over the Arabian Gulf and to evaluate diurnal variation.

*Data availability.* All AQABA data sets used in this study are permanently stored in an archive on the KEEPER service of the Max Planck Digital Library (https://keeper.mpdl.mpg.de; last access: 9 February 2021) and are available to all scientists, which agree to the AQABA data protocol.

*Competing interests.* The authors declare that they have no conflict of interest.

*Author contributions.* HF and DD designed the study. DD and BH performed the HCHO measurements. IT and UP provided the NO$_x$ measurements. RR, ST, MM and HH performed the LIF OH and HO$_2$ measurements during AQABA. EP and NW provided the OH reactivity measurements. NW, AE and CS were responsible and OVOC measurements. EB and LE carried out the VOC measurements during AQABA, which were used together with the OVOCs by EP to calculate speciated OH reactivity. JW supervised the VOC, OVOC and OH reactivity measurements. PE and JNC performed the O$_3$ and SO$_2$ measurements during AQABA. JS and JNC provided the actinic flux measurements and calculated photolysis rates. JL designed and supervised the AQABA project.

*Acknowledgements.* We thankfully acknowledge the cooperation with the Cyprus Institute (CyI), the King Abdullah University of Science and Technology (KAUST) and the Kuwait Institute for Scientific Research (KISR). We thank Hays Ships Ltd, Captain Pavel Kirzner and the *Kommandor Iona's* ship crew for the great support during all weather or wavy conditions and for an unforgettable time on board. We would like to thank especially Marcel Dorf and Claus Koeppel for the organization of the campaign and Hartwig Harder for the management on board. Last but not least we are grateful for the whole AQABA community and a successful campaign. EB, JW and JL additionally acknowledge the EMME-CARE project from the European Union's Horizon 2020 Research and Innovation Programme (grant agreement No. 856612), as well as matching co-funding by the Government of the Republic of Cyprus.

*Financial Support.* The article processing charges for this open-access publication were covered by the Max Planck Society.

*Review statement.*

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
