# Peer review of "Measurement report: Observation-based formaldehyde production rates and their relation to OH reactivity around the Arabian Peninsula"

_Atmospheric Chemistry and Physics, 2021_

## Author Comment (AC1)

**Referee 1:**

We would like to thank the referee for the detailed review of our manuscript and the highly valuable feedback. The referee's comments (black) our replies (blue) and the resulting modifications to the manuscript (red) are listed below.

*The author report measurements of HCHO on a ship cruise. Though this is a measurement report, the discussion is partly not very satisfying and it is not very clear, if there is any deeper meaning in the analysis. It remains unclear, if there is anything to learn from the HCHO yield that is calculated, specifically if the inorganic fraction of the OH reactivity is included. It may make it easier to exclude inorganic species that do not produce HCHO from the beginning.*

We modified a few figures and now used the OH reactivity corrected for non-HCHO yielding reactions for the main plots. The NOx dependence of the HCHO production is now discussed in more detail, although we decided to remove the scatter plot of $\alpha_{eff}$ vs $NO_x$, as a more detailed study would be necessary. In general, the dataset covers modestly polluted air masses with respect to $NO_x$ and thus we think it is likely that we almost never achieved conditions which favor the formation of peroxides through recombination of $RO_2$ instead of the production of HCHO through $RO_2+NO$.

One goal of the study was to check whether the calculation of the HCHO yield ($\alpha$) in different regions correlates with the variation of the measured OH reactivity towards different chemical families (VOCs, OVOCs, aromatics, alkenes) as a proof of concept. The results represent first studies of the HCHO dataset and thus we decided to write a measurement report to highlight the change in chemical composition during AQABA.

*Specifically, the dependence on NOx drastically change, because NO and NO2 are OH reactants, but what the authors want to point out is that the fate of RO2 and therefore the product distribution depends on the availability of NO. In addition, the authors want to connect the HCHO production with the chemistry and not just with the presence of OH reactants that do not produce HCHO (e.g. HCHO would be very low in an environment with high NO2 though there might be a lot of efficient HCHO from the chemistry of the organic compounds). The argument that subtracting the inorganic part reduces the data set is weak because the inorganic fraction reducing the chemical meaning of the derived yield. In the discussion the authors mention also the results, if the inorganic fraction is subtracted, but there is no meaningful interpretation.*

*P3 L2: "reactions" instead of "reaction"*

Correction made.

*P2/3: It is confusing that Equation 1 is defined for the sum of all reactants, but that later the authors distinguish between different type of OH reactants. This should be consistent specifically regarding the "yield".*

In order to clarify this, we have added new indices in Eq. 1 and improved the discussion of the results. The HCHO yield was calculated including all VOCs, we only excluded the contribution of inorganic trace gases and non-HCHO yielding reactions (R(OH)$_{eff}$).

P2 L30 – p3 L4:

In addition to summing up the contributions of individual HCHO production pathways, the production rate of HCHO resulting from reactions involving OH chemistry (P$_{OH}$(HCHO)) can be deduced from the

OH concentration ([OH]), the HCHO yield $\alpha_i$ and the OH reactivity (R(OH)) which represents the summation of all trace gases $R_i$ that react with OH with the rate coefficient $k_i$ (Liu et al. 2017; Wolfe et al., 2019):

$$P_{OH}(HCHO) = \alpha_i \cdot [OH] \cdot R(OH)_i \qquad (1)$$

with

$$R(OH)_i = \sum k_i \cdot R_i \qquad (2)$$

P4 L14–16:

For further interpretation, the effective HCHO yield $\alpha_{eff}$ was determined for each region by removal of non-HCHO yielding reactions (of NO, $NO_2$, $SO_2$, CO, HCHO and $O_3$ with OH) from the OH reactivity data (R(OH)$_{eff}$, Fig. S1).

$$\alpha_{eff} = \frac{P(HCHO) - P_{add}(HCHO)}{[OH] \cdot R(OH)_{eff}} \qquad (6)$$

*P3 Equation 4: Could transportation play a role?*

Transport of air masses e.g. by advection can have a significant influence on the HCHO mixing ratio, depending on the air mass origin. Since HCHO has many sources and is also released primarily, a strong influence is expected close to primary and secondary sources (e.g. ship plumes and close to cities). Obvious ship plumes were identified with $NO_x$, CO, $SO_2$ measurements and excluded from this study, as well as data during the night (sea breeze effect). We address the effects of transport later in the manuscript when interpreting the data and have also added the following at this point in the manuscript.

P3 L 31 – P4 L2:

The derived production and loss rates of HCHO can be influenced by direct emissions or by advective transport. Obvious direct emissions from ship plumes or other sources were excluded from the dataset. The potential role of transport is addressed in section 4.

*P4 Equation 5: It would be easier for the reader, if Eq. was written using the same terms as in the previous Equations.*

We changed $P_0$(HCHO) to $P_{add}$(HCHO) in Eq. 5. We also added additional information about $\alpha$, since it represents a lower estimate of the HCHO yield when including non-HCHO yielding reactions of OH. Equation 6 was added as the equivalent for the calculation of $\alpha_{eff}$ via R(OH)$_{eff}$.

P4 L 7–18:

Scatter plots of [OH] × R(OH) versus P(HCHO) yield the lower estimate of the formaldehyde yield $\alpha$ with respect to total OH chemistry (including $NO_x$, $SO_2$ and other non-HCHO yielding reactions, Fig. S2), reflecting the transition between rather clean to highly polluted conditions, both with respect to $NO_x$ and VOCs.

$$\alpha = \frac{P(HCHO) - P^*_{add}(HCHO)}{[OH] \cdot R(OH)} \qquad (5)$$

For further interpretation, the effective HCHO yield $\alpha_{eff}$ was determined for each region by removal of non-HCHO yielding reactions (of NO, $NO_2$, $SO_2$, CO, HCHO and $O_3$ with OH) from the OH reactivity data (R(OH)$_{eff}$, Fig. S1).

$$\alpha_{\text{eff}} = \frac{\text{P(HCHO)} - \text{P}_{\text{add}}\text{(HCHO)}}{\text{[OH]} \cdot \text{R(OH)}_{\text{eff}}}$$
(6)

*P7 L11: Are the assumptions about the photo-stationary state plausible, if maximum values were around noontime, when photooxidation is at its maximum and the chemical lifetime is 2.5h? Wouldn't the maximum expected to shift into the afternoon?*

The HCHO lifetime is sufficiently short that PSS is, to a good approximation, achieved around local noon. The hypothesis is supported by the fact that values of $d$ (HCHO) / dt were quite small on most days (Fig. 2), especially in the clean regions AS and MS. A shift to the afternoon can occur, when P(HCHO) is dominated by VOC oxidation processes which involve further photolysis steps / reactions prior to HCHO release.

Valin et al. 2016 (Fig. 2) also suggests PSS conditions during local noon with a HCHO lifetime of ~ 2.5 h, where L(HCHO) = P(HCHO).

*P8 L18: The reason for using of a constant value for the BHL instead of 30 min values is not plausible. Why does an interpolation of values lead to a lower data coverage compared to using a constant value?*

The (lack of) variability in the BLH and the associated uncertainty in this parameter do not warrant a more detailed treatment than use of a fixed BLH of 750 ± 113 m (± 1σ). We note additionally, that deposition accounts for only ~ 5–15 % of the HCHO loss, so use of an interpolated BLH would not change the results and/or conclusion of the analysis.

P9 L9–10:

For the BLH we used the mean value (± 1σ) of the ERA-5 results (750 ± 113 m).

*P9 L13: A compact linear relationship would only be expected, if chemical conditions do not change and transportation does not play a role. This should be made clear.*

We agree and have made this clear in the manuscript where we discuss the correlations between P(HCHO) and [OH]*R(OH) in detail.

P10 L3–6:

A compact linear relationship is expected under PSS conditions if chemical conditions do not change and the air mass is not affected by transport. While data points that do not fulfill PSS, (e.g. due to direct emissions from point sources, or advection of HCHO enriched or depleted air masses) are expected not to follow the regression line, yielding additional scatter.

*Figure 4: Error bars would help to see, how much of the scatter is explained by the statistical errors of data points.*

To maintain clarity of presentation, we included one representative data point with error bars for each plot.

*P11 Fig. 4:*

[Figure]

**Figure 4: Scatter plots with bivariate fits (York et al., 2004) of the product [OH] × R(OH)$_{eff}$ (± 58 %) versus the HCHO production rate P(HCHO) (± 62 %), subdivided into the different regions probed during the AQABA cruise. The slope of the respective regression represents the HCHO yield $\alpha_{eff}$, while the intercept can be interpreted as additional HCHO sources not related to OH chemistry (P$_{add}$(HCHO)).**

*P14 L12-221 and Figure 5: I do not see the meaning of plotting the yield against reactivity. Why should a yield linearly scale with the concentration of a family species? I would rather expect a constant number and a discussion, if the numbers are in any way expected from chemical oxidation mechanisms of these family species. In addition, the slopes in plots in Fig. 5 are essentially determined by one data point, because all other points are scattered in the low range. Please comment.*

The HCHO yield of a certain compound class is not expected to change under the same chemical conditions. To clarify this, we modified figure 5 and now use the relative contribution of the separated compound classes (R(OH)$_x$/R(OH)$_{eff}$) on the x-axis vs $\alpha_{eff}$, color-coded the z-axis with NO$_x$ and removed the bivariate linear fits. We now discuss in more detail how the increase in mixing ratio of a certain compound class would affect the HCHO yield per OH lost. The method used here is not feasible for a detailed study, since we cannot account for chemical aging and inhomogeneity in inflow with our dataset.

We also decided to exchange $\alpha$ for $\alpha_{eff}$ in the discussion of the manuscript (also in Table 1), since this adds more value when excluding inorganic and non-HCHO yielding reactions from the beginning. The extensive modifications made to the manuscript in response to these comments are listed below.

P11 L10–13:

For further analysis, we removed the highest contributors to inorganic reactivity (NO, NO$_2$, SO$_2$, O$_3$) and further non-HCHO yielding reactions (CO, HCHO) from R(OH), which resulted in so-called effective OH reactivity (R(OH)$_{eff}$). Even though the dataset was already filtered for stack emissions, NO$_x$ showed the highest contribution of these reactants (Fig. S1, Table S1), especially over the Gulf of Oman and while passing Bab-el-Mandeb (16.08.2017) and the street of Messina (30.08.2017).

P13 L14–15:

Removal of non-HCHO yielding reactions from the OH reactivity (R(OH)$_{eff}$) reduced the overall scatter significantly, which resulted in increased R² values for the Suez Canal and the Arabian Sea during leg 2 (Fig. 4, S2).

P14 L12–15:

However, the determined $\alpha_{eff}$ did not change significantly in most regions, which is also the case for the lowest values determined for the Arabian Sea (0.018) and the southern Red Sea (0.013). In these rather clean regions a rather low contribution of inorganic reactivity is expected, which was proven (Fig. S1).

P14 L21–32:

For further analysis, we have recalculated mean OH reactivities (± 1σ standard deviation) of the individual substance classes (R(OH)$_x$, e.g. alkanes, alkenes, OVOCs, aromatics) based on the findings of Pfannerstill et al. (2019). We assume that the unmeasured VOCs are proportional to the measured compounds of a certain compound class, so that these values represent general trends for the VOC oxidation of the subdivided regions (Table 1). Please note that we removed the reaction of HCHO with OH from the OVOC class, which is usually included when presenting speciated reactivity (Pfannerstill et al., 2019). The ratio of R(OH)$_x$/R(OH)$_{eff}$ then represents the relative contribution of a certain compound class to the regional so-called effective reactivity, which reflects the measured VOC oxidation plus unattributed reactivity. Alkanes (including CH$_4$) contributed on average 3 – 10 %, alkenes 2 – 10 %, OVOCs 8 – 33 % and aromatics 1 – 9 % to regional R(OH)$_{eff}$ (Table 1, Fig. 5). All sulfur-containing VOCs together contributed less than 1 % to the total R(OH)$_{eff}$ and were thus neglected. Elevated contribution of alkanes to R(OH)$_{eff}$ was found in SU and AG, while their contribution in the remaining regions was rather constant (~ 5 %) and dominated by the oxidation of methane.

P15 L2–5:

The sum of alkanes, alkenes, OVOCs and aromatic compounds on average contributed 19 % over the Mediterranean Sea, 32 % over the Suez Canal, 31 % over the northern Red Sea, 37 % over the southern Red Sea, 23 % over the Gulf of Aden, 16 % on the first leg over the Arabian Sea, 38 % over the Gulf of Oman, 62 % over the Arabian Gulf and 23 % on the second leg over the Arabian Sea to R(OH)$_{eff}$.

P15 Table 1: now includes $\alpha_{eff}$ instead of $\alpha$

P15 L13–23:

[revised manuscript text omitted]

*P16: In the discussion of the impact of NOx on the HCHO yield, the impact of NO as an OH reactant is missing, if the inorganic part is included. It is hard to see, if there is any meaning in the way the dependence is discussed. This part of the discussion needs significant improvement.*

The analysis has been modified with subtraction of the 'inorganic' reactions of OH (including NO, $NO_2$) from the beginning. The contribution of NO to the total OH reactivity was less than ~ 3 s$^{-1}$ in the used data set, this information is displayed in the SI (Fig. S1).

The discussion of the role of NO for the fate of $RO_2$ was added to the manuscript and we also added more information about the influence of NO for the recycling of $HO_x$. The $NO_x$ dependence should be clearer now, although we decided to remove fig. 6 form the manuscript (alpha against $NO_x$), since a more detailed study would be necessary to identify the role of $NO_x$ for HCHO production.

P17 L32 – P18 L10:

The yield of HCHO can also be interpreted as a function of $NO_x$ levels if there are processes (at low NO) that lead to formation of e.g. peroxides rather than HCHO (Wolfe et al., 2016), this occurs when the recombination of $CH_3O_2$ radicals is favoured instead of the reaction with NO to form the methoxy radical $CH_3O$ (Stickler et al., 2006). NO also directly influences the abundance of OH, as it recycles $HO_2$ radicals to OH (Whalley et al., 2010; Wolfe et al., 2016) and thus elevated $NO_x$ generally increases radical turnover. Valin et al. (2016) highlight that in isoprene rich regions, the influence of $NO_x$ on HCHO production is primarily due to its feedback on the production of OH. Wolfe et al. (2016) showed that the $NO_x$ dependence mainly affects the so-called prompt yield of isoprene oxidation as a non-linear function of $NO_x$. In general, the impact of $NO_x$ on HCHO production can be reduced to two factors: radical recycling and the termination of $RO_2$ radicals (Wolfe et al., 2016). The fact that very clean background MBL air was rarely sampled during AQABA (lowest median $NO_x$ of 0.16 ppbv over AS Leg 2) indicates that we rarely achieved conditions, in which the production of peroxides through the recombination of $CH_3O_2$ would dominate the production of HCHO through $CH_3O_2$ + NO. Tadic et al. (2020) also showed that most regions were in the $O_3$ production regime (signaling sufficient NO), which indicates that the production of HCHO was rather VOC and not $NO_x$ limited.

---

## Author Comment (AC2)

**Referee 2:**

We would like to thank the referee for the detailed review of our manuscript and the highly valuable feedback. The referee's comments (black) our replies (blue) and the resulting modifications to the manuscript (red) are listed below.

*The manuscript titled "Measurement report: Observation-based formaldehyde production rates and their relation to OH reactivity around the Arabian Peninsula" reported the shipborne observations of HCHO, OH, and OH reactivity from the Air Quality and Climate Change around in the Arabian Basin (AQABA) campaign in summer 2017. The authors calculated the effective yield of HCHO with total HCHO loss rates, OH and OH reactivity, by assuming a photochemical steady-state of HCHO and analyzed factors (e.g., NOx, VOC speciation, OH reactivity) that potentially impact the effective yield of HCHO. The measurements in the unique environment and the simultaneous measurements of HCHO, OH and OH reactivity will add value to the community but the analysis is not clear and needs major modifications. I'll recommend publication of this manuscript after the authors address my comments below.*

*Page 1 Line 28: Biogenic is the natural emission from living organisms such as plants and does not include biomass burning. Most HCHO from biogenic sources is secondary produced.*

We rewrote the introduction:

P1 L27 – P2 L3:

Formaldehyde (HCHO) is a ubiquitous trace gas that can help provide insight into the dynamical and chemical processes controlling atmospheric composition as an important source of hydroperoxyl radicals ($HO_2$) (Volkamer et al., 2010; Whalley et al., 2010; Anderson et al., 2017). The global atmospheric distribution of HCHO is dominated by in situ production during the oxidation of volatile organic compounds (VOCs) (Fortems-Cheiney et al., 2012; Anderson et al., 2017), although primary emissions from biomass burning (Akagi et al., 2011; Coggon et al., 2019; Kluge et al., 2020), vegetation (DiGangi et al., 2011), the industry sector (Parrish et al., 2012), shipping (Marbach et al., 2009; Celik et al., 2020) and agriculture (Kaiser et al., 2015) can contribute significantly to the local HCHO abundance.

*Page 3 eqn 3 and eqn 4: If you did not use eqn3 to calculate p(HCHO), please state it clearly especially when you provide the detailed information about each term. when you used L(HCHO) in eqn 4 to calculate P(HCHO), would the steady-state assumption hold at zero OH? It is not clear that the intercept can represent additional HCHO sources to VOC oxidation by OH and major question 1 can be addressed. The intercept looks like to be the dry deposition term actually. Please also provide more information about the uncertainties in the dry deposition term.*

Thanks a lot for this input, we hope the calculation is now explained more clearly in the manuscript.

In steady-state P(HCHO) = L(HCHO). For the assessment of PSS, we utilized scatter plots (fig.4), which represent the main loss and production processes in clean conditions. Scatter in these plots can be caused by direct emissions of HCHO, advection (transport) and due to deviation from photochemical steady-state (p 9 line 12-15). The additional scatter due to longer lifetime of HCHO (weak solar radiation) would be visible close to zero OH. Most of this data was removed anyway, since we only used daytime data. Some additional scatter is visible in e.g. AS Leg 2 and AG. In general, d(HCHO)/dt was quite small during AQABA as we experienced rather small diurnal variation of HCHO on most days (fig.2), which is also reflected in the total loss of HCHO (fig.3).

We compared the calculated deposition to the intercept b of the scatter plots (Fig. 4, P(HCHO) vs. R(OH)$_{eff}$ x [OH]):

| Region | Deposition (average) pptv s$^{-1}$ | Intercept b pptv s$^{-1}$ | Deposition / Intercept % |
|--------|-----------------------------------|---------------------------|--------------------------|
| AS1 | 0.0015 ± 0.0008 | 0.009 ± 0.002 | 16.6 |
| GO | 0.0088 ± 0.0046 | 0.058 ± 0.007 | 15.2 |
| AG | 0.0240 ± 0.0125 | 0.127 ± 0.050 | 18.9 |
| AS2 | 0.0052 ± 0.0027 | 0.035 ± 0.004 | 14.8 |
| GA | 0.0049 ± 0.0025 | 0.050 ± 0.013 | 9.8 |
| RS | 0.0016 ± 0.0008 | 0.014 ± 0.003 | 11.4 |
| RN | 0.0042 ± 0.0021 | 0.051 ± 0.006 | 8.2 |
| SU | 0.0084 ± 0.0044 | 0.073 ± 0.015 | 11.5 |
| MS | 0.0053 ± 0.0028 | 0.038 ± 0.004 | 13.9 |

Using literature values for the deposition velocity, we calculate that dry deposition can account for only 8–19 % of the intercept. The deposition velocity would have to be underestimated by a factor of five or more or the BLH would have to be overestimated by a similar factor (or a combination of both).

It is more plausible that the intercept at zero OH is due to additional sources of HCHO such as the photolysis of OVOCs, reactions involving O$_3$ or direct emissions.

P3 L22–25:

In this study we used the PSS assumption to calculate P(HCHO) via its loss reactions:

$$P(HCHO) = L(HCHO) = \left( k_{OH+HCHO} \cdot [OH] + j_{HCHO} + \frac{v_d}{BLH} \right) \cdot [HCHO] \qquad (4)$$

P10 L3–6:

A compact linear relationship is expected under PSS conditions if chemical conditions do not change and the air mass is not affected by transport. While data points that do not fulfill PSS, (e.g. due to sporadic emissions from point sources, or advection of HCHO enriched or depleted air masses) are expected not to follow the regression line, yielding additional scatter.

P13 L22 – P14 L2:

Besides individual data points that do not follow the regression line, the intercept b of the bivariate fit can be interpreted based on Eq. 5 either as an additional loss of HCHO (e.g. wash out, deposition) or as a region wide attribution of additional HCHO sources not related to OH chemistry (P$_{add}$(HCHO), Fig 4). As mentioned above, the effect of wash-out should be negligible, since we only experienced a short rain event on the 12.08.2017 during the night. Using literature values for the deposition velocity in this study (Stickler et al., 2007), we calculate that dry-deposition can account for only 8–19 % of the intercept. The deposition velocity would have to be underestimated by a factor of five or more or the BLH would have to be overestimated by a similar factor (or a combination of both). While the elevated values representing the Arabian Gulf were influenced by the pollution event encountered at the center of the Gulf, the other regions representing enhanced concentrations of air pollution (SU, GO, RN) also show significantly elevated intercepts within their uncertainties (0.051–0.127 pptv s$^{-1}$) compared to the relatively clean regions (AS, MS, RS), where we determined smaller intercepts b (0.009–0.038 pptv s$^{-1}$). Additional HCHO production via the ozonolysis of alkenes, indicated by elevated levels of O$_3$ (Tadic et al. 2020) and ethene (Bourtsoukidis et al., 2019), seems very likely in AG and SU. Elevated levels of OVOCs (Wang et al., 2020) and CH$_3$OOH (Dienhart et al., 2021) were also detected in these regions

and thus it is more plausible to use the intercept b as an indicator for additional HCHO production ($P_{add}$(HCHO)) not related to OH chemistry, although it seems not feasible to use for quantification.

*Page 3 line 18: Do you mean photochemical steady-state instead of photo-stationary state? I think photo-stationary state can mean recombination of the photolysis products to form HCHO. Please replace all "photo-stationary state" terms by "photochemical steady-state" throughout the manuscript.*

We replaced all photo-stationary state with photochemical steady-state.

*The lifetime of HCHO is 2.5 hrs or more in the study. What is the HCHO lifetime at sunset and sunrise? Are they short enough to assume to be in steady-state?*

To ensure no great deviation from PSS, only data for which the HCHO lifetime was < 10 hrs were used, although the main production and loss pathways of HCHO both depend on solar irradiation (photolysis and OH production, see also Valin et al., (2016) Fig. 2). In addition, as discussed above we used the scatter plots (Fig. 4) to ascertain whether the PSS assumption is valid.

*Page 4 eqn 5: Please define P0(HCHO)*

$P_0$(HCHO) = $P_{add}$(HCHO); this was a typo from a previous version of the manuscript, thank you.

P4 Eq. 5:

$$\alpha = \frac{P(HCHO) - P_{add}(HCHO)}{[OH] \cdot R(OH)} \tag{5}$$

*Page 8 line 17: No stickler et al. (2007) in the reference section*

Stickler, A., Fischer, H., Bozem, H., Gurk, C., Schiller, C., Martinez-Harder, M., Kubistin, D., Harder, H., Williams, J., Eerdekens, G., Yassaa, N., Ganzeveld, L., Sander, R., and Lelieveld, J.: Chemistry, transport and dry deposition of trace gases in the boundary layer over the tropical Atlantic Ocean and the Guyanas during the GABRIEL field campaign, Atmos. Chem. Phys., 7, 3933–3956, https://doi.org/10.5194/acp-7-3933-2007, 2007.

*Page 9 line 15-18: Is direct emission of HCHO very small compared to unsaturated hydrocarbons in the fresh pollution plume? Please specify the region and time when the fresh pollution plume occurred. The HCHO and ROH in AG leg seem to be high.*

Bourtsoukidis et al. 2019 detected the highest mixing ratios of ethene during the pollution event (fig. S13, mean AG 0.788 ppb). At the same time the HCHO mixing ratios approached 12 ppb and $O_3$ (Tadic et al. 2020), acetaldehyde & acetone (Wang et al 2020) were enhanced. As already indicated in the text, we cannot differentiate between direct HCHO emissions, and HCHO that was formed as the plume was processed during transport from Kuwait / the western coast of the Gulf. The effect of the pollution plume is further discussed in section 4.

P10 L9-11:

The dataset covers a highly polluted event in AG (30.07.2017), although the high HCHO / R(OH) ratio, as well as enhanced Methanol, Acetaldehyde and $O_3$ mixing ratios (Wang et al., 2020; Tadic et al., 2020) indicate rather processed air masses which could also include primary emissions.

*Page 9 line 18-20: Specify which region and time had rainout and which region and time had primary emission.*

As already mentioned at page 3 line 30-31 no significant rain events occurred for this study during the AQABA campaign, only a minor precipitation event occurred in AS during night from 11. - 12.08.2017.

The primary emission of HCHO cannot be accurately identified, but we removed obvious pollution events (e.g. in plumes from ships or oil rigs) via co-variance with elevated $NO_x$, CO and $SO_2$ mixing ratios.

P10 L9-17:

The dataset covers a highly polluted event in AG (30.07.2017), although the high HCHO / R(OH) ratio, as well as enhanced Methanol, Acetaldehyde and $O_3$ mixing ratios (Wang et al., 2020; Tadic et al., 2020) indicate rather processed air masses which could also include primary emissions. Primary emission from point sources would likely shift data above the regression line due to the correlation with the HCHO mixing ratio (although this effect can be compensated depending on the OH concentration in emission plumes). The primary emission of HCHO cannot be accurately identified, but we removed obvious pollution events (e.g. in plumes from ships or oil rigs) via co-variance with with elevated $NO_x$, CO and $SO_2$ mixing ratios. Furthermore, rainout can affect the relationship, because it would result in data points shifted below the regression line, although this effect can be neglected in this study as only minor precipitation occurred in AS on the 12.08.2017 during night.

*Page9 line20- page 10 line 1-2: The intercept may not represent additional HCHO sources because the steady-state assumption does not work at very long lifetime. The intercept can be equal to the dry deposition term.*

The intercept (OH = 0) is the result of several processes including additional chemical loss and production, dry-deposition and may also be related to partial break-down of the PSS-assumption when OH is low (e.g. early morning, late evening). The intercept should thus not be overinterpreted and used to e.g. calculate deposition velocities. We only used it as in indicator for the possibility of additional HCHO sources, which is supported by covariance of the intercept with elevated levels of air pollution.

P10 L35 – P11 L4:

Non-zero P(HCHO) at zero [OH] × R(OH), corresponding to the intercept b of the linear regression, can be interpreted as the result of several processes including additional chemical loss and production, dry deposition and may also be related to partial break-down of the PSS assumption when OH is low (e.g. early morning, late evening). The intercept should thus not be overinterpreted and used to e.g. calculate deposition velocities.

*Page 10 line 9: How do primary emissions and transport affect the alpha derived here?*

Primary emissions of HCHO would result in data shifted towards the y-axis and thus may be partially responsible for the positive intercept on the y-axis. This would also bias the regression analysis and impact on $\alpha$. Primary emissions of HCHO at low OH would tend to increase the intercept, whereas primary emissions at high OH would bias $\alpha$ to larger values.

P10 L24-31:

The highly elevated P(HCHO) values over the Arabian Gulf (Fig. S2) also include data during a pollution event on the 30.07.2017 in the center of the Gulf (Fig. 2, see also Tadic et al. 2020; Wang et

al. 2020; Pfannerstill et al 2019; Bourtsoukidis et al., 2019). The evaluated $\alpha$ in AG is thus likely influenced by primary emissions or transport of air pollution from the western coast of the Gulf. However, this pollution event seems to be representative for this unique region and thus we did not exclude the data from the study. Primary emissions of HCHO would result in data shifted towards the y-axis and thus may be partially responsible for the positive intercept on the y-axis. This would also bias the regression analysis and impact on $\alpha$. Primary emissions of HCHO at low OH would tend to increase the intercept, whereas primary emissions at high OH would bias $\alpha$ to larger values.

*Page 11 line 12-17 Can primary emission of HCHO also play a role in the highest calculated pHCHO on 30/07/2017 from figure 2?*

Yes, as indicated above, primary emissions can bias the derivation of P(HCHO). We decided to keep the data from 30.07.2017 where we encountered air masses in the center of the Arabian Gulf, which included high mixing ratios of HCHO precursors e.g. acetaldehyde, methanol, acetone, ethene etc. $NO_x$ mixing ratios were below 8 ppb and thus while the highly elevated HCHO mixing ratios are a result of photochemical formation, primary emissions can not be ruled out. We added additional information that this data was likely influenced by air pollution transported from the western coast of the Gulf.

P12 L11–13:

P(HCHO) varied between 0 and approximately 1.2 pptv s$^{-1}$, with values above 0.4 pptv s$^{-1}$ only detected over the Arabian Gulf, which were likely influenced by air pollution transported from the west coast of the Gulf and Kuwait, where oil and gas industries are numerous (Pfannerstill et al., 2019).

*Page 12 line 20: which are these three regions mentioned? Please be specific.*

Removed this sentence from the manuscript and added additional information about wash out of HCHO.

P13 L22–25:

Besides individual data points that do not follow the regression line, the intercept b of the bivariate fit can be interpreted based on Eq. 5 either as an additional loss of HCHO (e.g. wash out, deposition) or as a region wide attribution of additional HCHO sources not related to OH chemistry ($P_{add}$(HCHO), Fig 4). As mentioned above, the effect of wash out should be negligible, since we only experienced a short rain event on the 12.08.2017 during the night.

*Page 12: As mentioned before, the intercept likely cannot be interpreted as Padd(HCHO) because of the long lifetime at OH=0. Do the intercepts correlate with dry deposition?*

See reply to comment above in which the contributions to the intercept are described in more detail.

*Page 13 line 13-23. "individual trace gases (R(OH)x) from alkanes (R(OH)alkanes), alkenes (R(OH)alkenes), OVOCs without HCHO" is confusing. Does x represent individual group (e.g., alkanes) instead of individual trace gas? I would expect less VOC species and higher measured R(OH)x to total R(OH) ratio in cleaner or lower ROH environments. No information about the VOCs measurement methods is provided in Section 2. A table of the measured VOC species is needed to understand R(OH)x/R(OH).*

Yes, R(OH)$_x$ represents the so-called 'speciated OH reactivity' towards a certain substance class (alkanes, alkenes, OVOCs, aromatics), which were calculated based on the findings of Pfannerstill et al. (2019), although we separated the Arabian Sea in 2 legs. Here we used it to sum up the contributions

towards certain compound classes to identify which regions covered a higher relative contribution of e.g. alkenes (Table 1, Pfannerstill et al. 2019).

We changed p13 line 13-15 to "individual substance classes ($R(OH)_x$, e.g. alkanes, alkenes, OVOCs, aromatics) based on the findings of Pfannerstill et al. (2019). We assume that the unmeasured VOCs are proportional to the measured compounds of a certain compound class, so that these values represent general trends for the VOC oxidation of the subdivided regions."

We also added the VOC measurements to section 2 and the measured VOC species are listed in table S2.

P6 L7–13:

Non-methane hydrocarbons (NMHCs) were measured with two coupled GC–FID systems (GC5000VOC and GC5000BTX; AMA Instruments GmbH, Germany). The GC5000VOC was used for the quantification of light hydrocarbons (C2–C6), while the CG5000BTX was used for the heavier hydrocarbons and aromatics (C6–C8). The NMHCs have average detection limits in the range of $1 - 25$ pptv and a total uncertainty in the range of $6 - 13$ % (Bourtsoukidis et al., 2019).

OVOCs were detected with a PTR-ToF-MS (8000, Ionicon Analytik GmbH, Innsbruck, Austria) with a total measurement uncertainty in the range of $6 - 17$ %, an accuracy of ~ 50 % and $3\sigma$ detection limits (derived by background measurements) of $52 \pm 26$ pptv for acetaldehyde, $22 \pm 9$ pptv for acetone and $9 \pm 6$ pptv for methyl ethyl ketone (Wang et al., 2020).

*Figure 5: Do you assume the unmeasured VOC groups are proportional to the measured R(OH)alkene, R(OH)alkanes, R(OH)OVOCs, and R(OH)aromatics? What about using the ratio R(OH)x / ROH on the x axis? This sounds the way to go if you want to evaluate the impact of the different VOC groups on HCHO yield alpha.*

Thank you for this advice, we changed the x-axis to $R(OH)_x / R(OH)_{eff}$. We also color-coded the z-axis to identify certain regions during AQABA more easily and to check whether a trend with elevated $NO_x$ is visible.

P14 L21–28:

For further analysis, we have recalculated mean OH reactivities ($\pm 1\sigma$ standard deviation) of the individual substance classes ($R(OH)_x$, e.g. alkanes, alkenes, OVOCs, aromatics) based on the findings of Pfannerstill et al. (2019). We assume that the unmeasured VOCs are proportional to the measured compounds of a certain compound class, so that these values represent general trends for the VOC oxidation of the subdivided regions (Table 1). Please note that we removed the reaction of HCHO with OH from the OVOC class, which is usually included when presenting speciated reactivity (see Table 1 in Pfannerstill et al., 2019). The ratio of $R(OH)_x/R(OH)_{eff}$ then represents the relative contribution of a certain compound class to the regional so-called effective reactivity, which reflects the measured VOC oxidation plus unattributed reactivity.

*Page 15 line 19-page 16 line 1-2: I think the statement "the role of NOx in HCHO production is ambiguous" does not sound correct based on Wolfe et al. (2016) and Valin et al., (2016). Do you mean HCHO yield instead? The following explanation is also not clear. It is more evident that NO influencing the amount of OH radical can impact the production of HCHO. Explain how NO influencing the amount of OH radical can impact the yield of HCHO.*

We improved the discussion about the $NO_x$ dependence of the HCHO production by including the $HO_x$ recycling and discuss the importance of the fate of $RO_2$ radicals. We also decided to remove fig. 6 from the manuscript, as the AQABA dataset mainly covers modest polluted air masses, covering sufficient levels of NO to induce HCHO instead of peroxide formation. Thus the HCHO

production was more likely limited by the abundance of oxidants and reactive VOCs rather than $NO_x$.

P17 L32 – P18 L10:

The yield of HCHO can also be interpreted as a function of $NO_x$ levels if there are processes (at low NO) that lead to formation of e.g. peroxides rather than HCHO (Wolfe et al., 2016), this occurs when the recombination of $CH_3O_2$ radicals is prioritized instead of the reaction with NO to form the methoxy radical $CH_3O$ (Stickler et al., 2006). NO also directly influences the abundance of OH, as it recycles $HO_2$ radicals to OH (Whalley et al., 2010; Wolfe et al., 2016) and thus elevated $NO_x$ generally increases radical turnover. Valin et al. (2016) found that in isoprene rich regions, the influence of $NO_x$ on HCHO production is primarily due to its feedback on the production of OH. Wolfe et al. (2016) showed that the $NO_x$ dependence mainly affects the so-called prompt yield of isoprene oxidation as a non-linear function of $NO_x$. In general, the impact of $NO_x$ on HCHO production can be reduced to two factors: radical recycling and the termination of $RO_2$ radicals (Wolfe et al., 2016). The fact that very clean background MBL air was rarely sampled during AQABA (lowest median $NO_x$ of 0.16 ppbv over AS Leg 2) indicates that we rarely achieved conditions, in which the production of peroxides through the recombination of $CH_3O_2$ would dominate the production of HCHO through $CH_3O_2$ + NO. Tadic et al. (2020) also showed that most regions were in the $O_3$ production regime (signaling sufficient NO), which indicates that the production of HCHO was rather VOC and not $NO_x$ limited.

*Page 17 line 10-11: again "photochemical steady-state" instead of "photo-stationary state".*

Correction made.

*Page 17 Line 11-12: Could you provide potential reasons why the yield of HCHO depends largely on the presence of absolute VOCs levels? This does not have a clear physical meaning to me.*

You are right, we removed this part.

P18 L25-27:

The region-wide yield of HCHO ($\alpha_{eff}$) from reactions of OH differs along the various chemical regimes encountered with lowest values (less than 2 %) deduced over the Arabian Sea during the first leg and the southern Red Sea, and highest values (~ 32 %) over the Arabian Gulf.

---

## Author Response (AR2)

**Author's response**

We would like to thank John Orlando for the management of our manuscript. The author's comments (black), the referee's comments (grey) and our replies (blue) are listed below. All resulting changes are highlighted in the new version of the manuscript, including linguistic corrections.

**Editor:**
Thank you for re-submitting your manuscript. As you are aware, the reviewers have made further (minor) suggestions, as listed below.

- Page 8 line 8 should be $4.5 \times 10\text{-}5$ and $7.8 \times 10\text{-}5$?

Correction made.

- According to eqn 4 and 5, when [OH] R(OH) is equal zero, should the loss term J+k[OH] also be zero? It looks to me that the intercept in figure 4 is basically from the dry deposition term.

I believe that you have addressed the issue of the intercept in Figure 4 already, but please have another look in case more can be done to clarify.

We agree that the loss terms j(HCHO) and $k_{HCHO+OH}$ are basically zero when OH $\times$ R(OH) equals zero, thus, the intercept can be interpreted as additional loss e.g. due to dry deposition (Eq. 4 & 5) or as additional source of HCHO based on the PSS assumption.

We would like to highlight, as already addressed in the response to Referee 2, that we included the dry deposition based on literature values and ERA5 results. Based on this calculation the dry deposition only accounts for 8-19% of the intercept. Although we cannot exclude the underestimation of the dry deposition, we think it is more appropriate to use the intercept as an indicator for additional HCHO sources (not necessarily related to OH chemistry). The interference of other primary HCHO sources was likely during the measurements around the Arabian Peninsula and high concentrations of $O_3$ and unsaturated hydrocarbons were detected in the Arabian Gulf. We address this issue more clearly on p13 line 19-30.

In addition, we improved the discussion of the results in the new version of the manuscript.